# MITF reprograms the extracellular matrix and focal adhesion in melanoma

**Ramile Dilshat[1], Valerie Fock[1], Colin Kenny[2], Ilse Gerritsen[1], Romain Maurice Jacques Lasseur[1], Jana Travnickova [3], Ossia M Eichhoff [4], Philipp Cerny[1], Katrin Möller[1†], Sara Sigurbjörnsdóttir [1], Kritika Kirty[1], Berglind Ósk Einarsdottir[1], Phil F Cheng[4], Mitchell Levesque[4], Robert A Cornell[2], E Elizabeth Patton[3], Lionel Larue[5], Marie de Tayrac [6,7], Erna Magnúsdóttir [8], Margrét Helga Ögmundsdóttir[1], Eirikur Steingrimsson[1]\***

[1]Department of Biochemistry and Molecular Biology, BioMedical Center, Faculty of Medicine, University of Iceland, Reykjavik, Iceland; [2]Department of Anatomy and Cell biology, Carver College of Medicine, University of Iowa, Iowa City, United States; [3]MRC Institute of Genetics and Molecular Medicine, MRC Human Genetics Unit, University of Edinburgh, Edinburgh, United Kingdom; [4]Department of Dermatology, University Hospital Zurich, Zurich, Switzerland; [5]Institut Curie, CNRS UMR3347, INSERM U1021, Centre Universitaire, Orsay, France; [6]Service de Génétique Moléculaire et Génomique, CHU, Rennes, France; [7]Univ Rennes1, CNRS, IGDR (Institut de Génétique et Développement de Rennes), Rennes, France; [8]Department of Anatomy, BioMedical Center, Faculty of Medicine, University of Iceland, Reykjavik, Iceland

**\*For correspondence:**
eirikurs@hi.is

**Present address:** [†]Institute of Molecular Life Sciences, University of Zurich, Zurich, Switzerland

**Competing interests:** The authors declare that no competing interests exist.

**Abstract** The microphthalmia-associated transcription factor (MITF) is a critical regulator of melanocyte development and differentiation. It also plays an important role in melanoma where it has been described as a molecular rheostat that, depending on activity levels, allows reversible switching between different cellular states. Here, we show that MITF directly represses the expression of genes associated with the extracellular matrix (ECM) and focal adhesion pathways in human melanoma cells as well as of regulators of epithelial-to-mesenchymal transition (EMT) such as CDH2, thus affecting cell morphology and cell-matrix interactions. Importantly, we show that these effects of MITF are reversible, as expected from the rheostat model. The number of focal adhesion points increased upon MITF knockdown, a feature observed in drug-resistant melanomas. Cells lacking MITF are similar to the cells of minimal residual disease observed in both human and zebrafish melanomas. Our results suggest that MITF plays a critical role as a repressor of gene expression and is actively involved in shaping the microenvironment of melanoma cells in a cell-autonomous manner.

## Introduction

Melanoma is a highly aggressive form of skin cancer that originates from melanocytes. Approximately, 60% of melanoma tumors harbor a BRAF mutation, most often BRAF$^{V600E}$, which leads to hyperactivation of the mitogen-activated protein kinase (MAPK) pathway (*Davies et al., 2002*). Drugs targeting the BRAF and MAPK pathways are clinically important, but almost invariably, resistance arises within a short time period (*Kugel and Aplin, 2014*). Melanoma inherits its aggressive nature from its multipotent neural crest precursors that gives rise to various cells including melanocytes, glia, and adrenal cells (*Le Douarin and Kalcheim, 1999*; *Le Douarin and Dupin, 2018*). The developmental programme of neural crest cells is believed to be reinitiated during melanoma

progression and dysregulation of neural crest genes is predictive of metastatic potential and negative prognosis in melanoma (*Mascarenhas et al., 2010*; *Bailey et al., 2012*; *Kulesa et al., 2006*). Various different studies, including gene expression studies of tumors, immunohistochemical analysis of melanoma samples and single-cell sequencing studies of patient-derived xenografts suggest the existence of different cell types in melanoma tumors. This cellular heterogeneity is believed to reflect the associated ability of tumor cells to switch their phenotype from proliferative, non-invasive cells to quiescent, invasive cells and back, thus allowing metastasis and the escape from therapeutic intervention (reviewed in *Rambow et al., 2019*). This has been summarized in the phenotype switching model which suggests that melanoma cells can switch between invasive and proliferative states allowing them to either grow and form tumor or metastasize to a new site (*Rambow et al., 2019*; *Hoek and Goding, 2010*). Understanding the molecular mechanisms underlying the phenotypic plasticity of melanoma cells is key to addressing the metastatic potential of melanoma cells.

The microphthalmia-associated transcription factor (MITF) is essential for melanocyte differentiation, proliferation, and survival. MITF is also important during melanomagenesis (reviewed in *Goding and Arnheiter, 2019*). This is best evidenced by the observations that the rare germline mutation E318K of MITF increases the susceptibility to melanoma and MITF has been shown to be amplified in 15% of melanoma tumors (*Bertolotto et al., 2011*; *Garraway et al., 2005*; *Yokoyama et al., 2011*). Importantly, MITF activity has been used as a proxy for the phenotype switching model with MITF$^{high}$ cells characterized as proliferative, whereas MITF$^{low}$ cells have been assigned a quiescent invasive phenotype (*Carreira et al., 2006*; *Hoek et al., 2006*). In fact, MITF has been proposed to act as a rheostat where the levels of MITF activity determine the phenotypic state of melanoma cells (reviewed in *Rambow et al., 2019*). Since MITF expression and activity are regulated by the various signaling pathways, the tumor microenvironment has been proposed to instruct phenotypic changes in melanoma cells and thus foster disease progression (*Feige et al., 2011*; *Miskolczi et al., 2018*; *Widmer et al., 2013*; *Riesenberg et al., 2015*). However, antibody staining suggests that cells lacking MITF are abundant in melanomas (*Goodall et al., 2008*) and single-cell sequencing of human xenotransplants and of zebrafish melanoma models suggest the existence of cells with very low MITF expression (*Rambow et al., 2018*; *Travnickova et al., 2019*). These cells belong to a population of cells believed to represent minimal residual disease, cells that remain viable upon drug exposure.

The extracellular matrix (ECM) is an important component of the tumor microenvironment as it provides cells with biochemical and structural support. In melanoma, expression of ECM proteins such as tenascin and fibronectin increases during disease progression (*Frey et al., 2011*). Focal adhesions not only offer physical attachment of cells to the ECM through the integrin receptor, but also initiate signaling cascades that regulate cell proliferation, migration, and survival (*Mitra et al., 2005*; *Geiger et al., 2001*; *Playford and Schaller, 2004*). A key focal adhesion signaling protein is Focal Adhesion Kinase (FAK), which activates the ERK pathway via Grb-FAK interactions (*Schlaepfer et al., 1999*). An important scaffolding protein at the focal adhesion complex is Paxillin (PXN) which recruits other proteins to the focal adhesion sites when phosphorylated by FAK and SRC (*Deakin and Turner, 2008*). Importantly, phosphorylation of PXN is critical for activation of RAF, MEK, and ERK and has been shown to confer drug resistance by activating Bcl-2 through ERK signaling (*Wu et al., 2016*; *Sen et al., 2010*; *Ishibe et al., 2003*; *Sen et al., 2012*; *Hirata et al., 2015*). This highlights the importance of identifying a molecular mechanism that confers cells with the ability to circumvent drug inhibition through phenotypic changes.

In this study, we show that MITF represses the expression of focal adhesion and ECM genes in melanoma cells and tissues. Our findings reveal a new role for MITF in regulating the expression of genes that are essential for creating the melanoma microenvironment, establishing a link to melanoma progression and drug resistance.

## Results

### Melanoma cells devoid of MITF are enlarged and exhibit altered matrix interactions

To assess the effects of permanent loss of MITF in melanoma cells, we used the clustered regularly interspaced short palindromic repeats (CRISPR)-Cas9 technique to generate MITF knockout (KO) cell

lines in the human hypo-tetraploid SkMel28 melanoma cell line (containing four copies of MITF). We targeted exons 2 (an early exon containing a transactivation domain) and 6 (containing the DNA-binding domain) of MITF separately and the resulting isogenic cell lines are hereafter referred to as ΔMITF-X2 and ΔMITF-X6 (*Figure 1a*). The control cell line EV-SkMel28 was generated by transfecting SkMel28 cells with Cas9 along with the empty gRNA plasmid. To identify mutations introduced in the cell lines, we performed whole genome sequence (WGS) analysis, which showed that mutations were introduced in MITF in both the ΔMITF-X2 and ΔMITF-X6 cells (*Figure 1b,c*) but not in the EV-SkMel28 control. In addition, we confirmed the WGS analysis by amplifying the mutated genomic regions, cloning them into vectors and performing Sanger sequencing. The ΔMITF-X2 line had two different but independent insertion mutations in the same codon (insertion of A and T in the codon for Y22) and a 5 bp deletion (encoding Y22 and H23) that are present in 64%, 19%, and 17% of sequenced DNA fragments in this region, respectively. All these mutations introduced frameshifts and premature stop codons in exon 2 of MITF (*Figure 1b*). Sanger sequencing of DNA clones containing PCR amplified cDNA fragments from the MITF-gene of ΔMITF-X2 cells verified that the ΔMITF-X2 cells have the same mutations at similar frequency as observed in the WGS data (*Figure 1—figure supplement 1c*). The mutations present in the ΔMITF-X6 line are the following: 52% of the sequenced fragments contained a deletion of 1 bp (encoding residue A198), 33% contained a 6 bp in-frame deletion in the basic domain of the protein (encoding residues R197-R198), and 15% of the sequenced fragments contained a 17 bp deletion (encoding residues 198–203). Both the 1- and 17 bp deletions introduced frameshifts and downstream stop codons (*Figure 1c*), whereas the in-frame 6 bp deletion removed two amino acids at the beginning of the alpha-helix encoding the basic domain and is therefore not expected to be able to bind DNA. No wild-type MITF gene was detected in either cell line. In both cell lines, the ratio of mutants is consistent with two chromosomes carrying the same mutation and the remaining two chromosomes each carrying a different mutation. Western blotting revealed that the ΔMITF-X6 cells express very little, if any, MITF protein. Although the ΔMITF-X2 cells did not express the full-length ~55 kDa MITF protein, truncated forms of MITF were detected at ~40 and 47 kDa (*Figure 1d*). These truncated forms were also present in wild-type cells, albeit at lower levels, suggesting that these are alternative isoforms of the MITF protein (*Figure 1d*). In order to determine if these shorter isoforms are due to alternative splicing, we performed RT-PCR across several exon-intron borders around exon 2 of the MITF transcript. Our results did not show any alternative splice forms of MITF (*Figure 1—figure supplement 1a–b*). Similarly, neither the WGS nor RT-PCR studies showed evidence for transcripts lacking exon2 from the ΔMITF-X2 cDNA indicating that the truncated MITF proteins are most likely products of alternative translation start sites (*Figure 1—figure supplement 1c*). The C5 MITF antibody used here recognizes an epitope located between residues 120 and 170 of MITF, which corresponds to exons 4 and 5 (*Figure 1a*; *Fock et al., 2019*). The truncated proteins observed in wild type and ΔMITF-X2 cells must still contain this region and are therefore likely to arise from alternative translation start sites. Immunostaining revealed a mostly nuclear staining of MITF in both the EV-SKmel28 and ΔMITF-X2 cells (*Figure 1e*), indicating that the truncated MITF isoforms reside in the nucleus. However, in the ΔMITF-X6 cells, no signal for MITF was observed in the nucleus, whereas a very low background signal was observed in the cytoplasm (*Figure 1e*). To summarize, we have generated the ΔMITF-X6 cells that are devoid of wild type MITF and the ΔMITF-X2 cells that carry a hypomorphic mutation. Below, we refer to both as CRISPR MITF-KO cell lines due to the way they were generated.

Morphological analysis revealed that both MITF-KO cell lines exhibited enlarged cytoplasm as compared to controls (*Figure 1e–g*). Vimentin staining revealed enlarged cells (*Figure 1e*), which is consistent with a report showing that loss of MITF affects the cytoskeletal structure and shape of melanoma cells (*Carreira et al., 2006*). Quantification of phase-contrast microscopy images revealed that the average size of the ΔMITF-X2 and ΔMITF-X6 cells was 1.7-fold larger than the EV-SkMel28 cells (*Figure 1g*). To characterize the behavior of the cell lines when provided with ECM that mimics the basement membrane, we seeded the cells on top of matrigel-coated slides, supplemented with complete growth medium containing 2% matrigel. Both MITF-KO cell lines formed aggregates, whereas the control EV-SkMel28 cells displayed a flat sheet-like morphology (*Figure 1h*). Taken together, our results show that loss of MITF lead to changes in cell morphology and cell-matrix interactions.

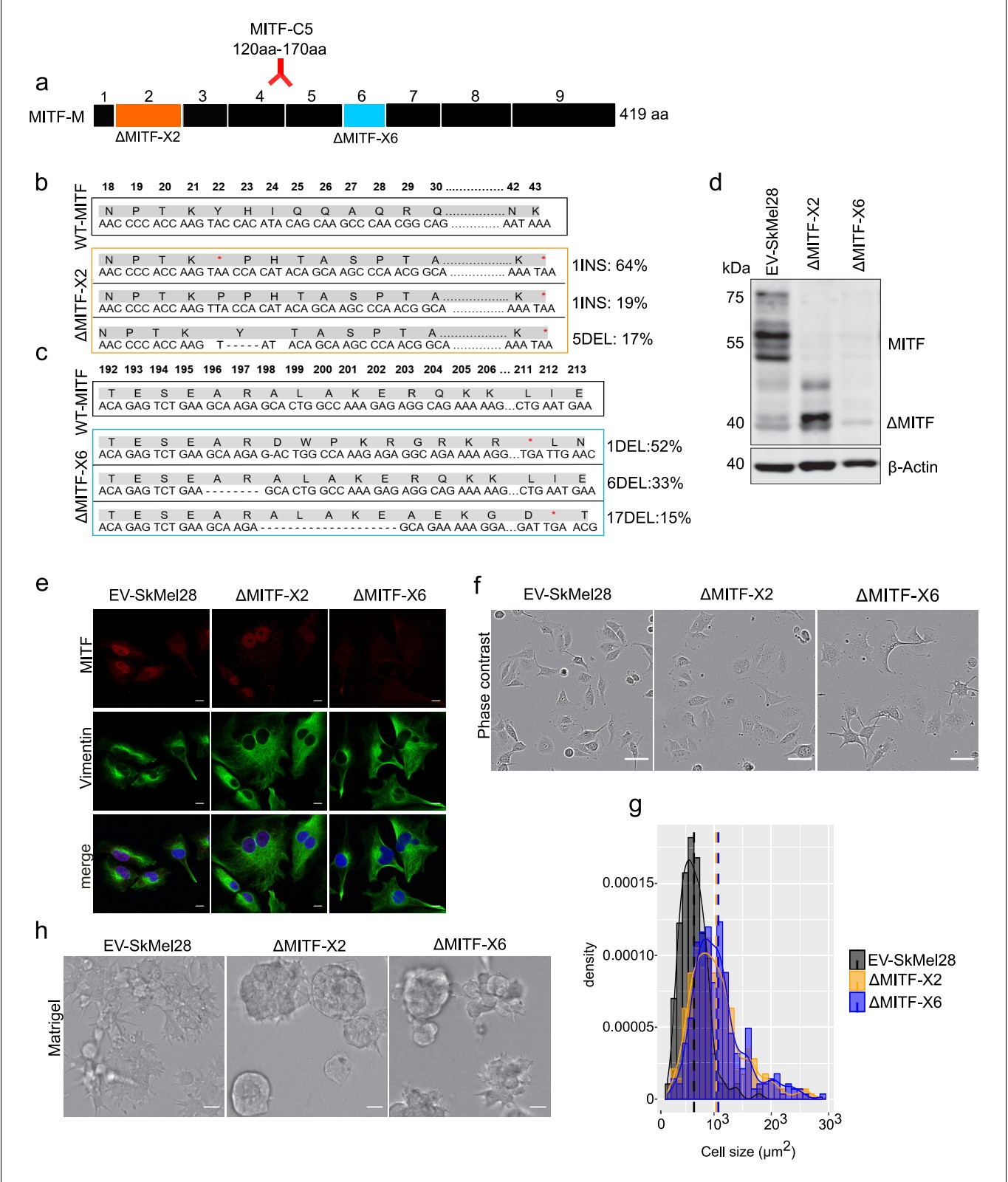

**Figure 1.** MITF depletion affects cell size and cell-matrix interaction. (a) Schematic illustration of MITF-M isoform and gRNA targeted location at exon 2 and exon 6. The epitope location for MITF C5 antibody spanning exons 4 and 5 is shown. (b, c) Mutations detected in ΔMITF-X2 and ΔMITF-X6 cell lines; amino acid sequence numbering was indexed relative to MITF-M. Percentage of mutations was derived from WGS analysis by counting sequenced fragments aligned to the mutated regions. (d) Western blot showing the MITF band in EV-SkMel28, ΔMITF-X2, and ΔMITF-X6 cell lines. (e)

*Figure 1 continued on next page*

Figure 1 continued

Immunostaining for MITF and Vimentin in EV-SkMel28, ΔMITF-X2, and ΔMITF-X6 cell lines, scale bar (10 μm). (f, g) Phase-contrast microscopy and cell size quantification using Image J with at least 200 images taken for both MITF-KO and EV-SkMel28 cell lines, scale bar (10 μm). Average cell size for each cell line is indicated by dashed lines; EV-SkMel28 cells (6502 μm$^2$, SEM: 460), ΔMITF-X2 (10,395 μm$^2$, SEM: 270) and the ΔMITF-X6 (10,825 μm$^2$, SEM: 330). (h) Bright-field images of MITF-KO and EV-SkMel28 cells grown on top of matrigel, scale bar (10 μm).

The online version of this article includes the following source data and figure supplement(s) for figure 1:

**Source data 1.** Cell size measurement of MIT-KO cells.
**Figure supplement 1.** Analysis of MITF mRNA and protein in MITF-KO cells.

## Expression of ECM and focal adhesion genes is increased upon loss of MITF

Next, we compared the transcriptomic profile of the ΔMITF-X6 cells (exhibiting complete loss of wild type MITF) to the EV-SkMel28 control cells. We identified 2136 differentially expressed genes (DEGs) between ΔMITF-X6 and EV-SkMel28 cells with the cut off qval <0.05 (*Supplementary file 1*). Of these, 1516 genes showed twofold change in expression (*Figure 2a*). Gene ontology and KEGG pathway enrichment analysis revealed that the genes reduced in expression upon MITF depletion were verified MITF-target genes involved in pigmentation and pigment cell differentiation such as *DCT*, *MLANA*, *OCA2*, and *IRF4* in addition to *MITF* itself (*Figure 2a,b*, *Supplementary file 1*). Genes whose expression was increased upon loss of MITF were enriched in processes involved in glycosaminoglycan metabolism, ECM organization and extracellular structure organization, and included genes such as *SERPINA3*, *ITGA2*, *PXDN*, and *TGFβ1* (*Figure 2a,b*, *Supplementary file 1*).

As MITF is central to the melanoma phenotype switching model (*Hoek et al., 2008*), we were interested whether loss of MITF would be consistent with the published transcriptional signatures linked to phenotype switching in melanoma cells (*Verfaillie et al., 2015*). Enrichment analysis (*Yu et al., 2012*) showed that MITF-dictated transcriptional signatures such as 'Hoek proliferative' (*Hoek et al., 2006*), 'Tsoi melanocytic' (*Tsoi et al., 2018*), 'Jonsson pigmentation' (*Jönsson et al., 2010*), and 'Tirosh MITF program' (*Tirosh et al., 2016*) were reduced in expression in the ΔMITF-X6 cells, whereas MITF negative gene signatures including 'Tsoi neural crest like' (*Tsoi et al., 2018*), 'Hoek invasive' (*Hoek et al., 2006*), 'Tirosh AXL program' (*Tirosh et al., 2016*), and 'Rambow NCSC program' (*Rambow et al., 2018*) were induced in the ΔMITF-X6 cells (*Figure 2c*). This further validated the MITF-KO cells as a representative model of long-term MITF loss.

In order to investigate if the genes affected by MITF loss are direct targets of MITF, we used the CUT-and-RUN method to map protein-DNA interactions (*Skene and Henikoff, 2017*). Briefly, a chromatin isolate was incubated with an antibody against MITF and then Protein A/G fused with Micrococcal nuclease (MNase) was added to cut the DNA that is recognized by the target antibody. The resulting DNA fragments were then sequenced (*Skene and Henikoff, 2017*). We used an anti-MITF antibody (i.e. MITF CUT-and-RUN) in the SkMel28 cells to map MITF genome-wide binding sites (*Meers et al., 2019*). We identified 37,643 peaks located 10 kb +/- from the TSS, 3'UTR or intronic regions of 8288 genes (*Figure 2—figure supplement 1a*; *Supplementary file 2*). Gene ontology analysis revealed that among genes associated with MITF CUT-and-RUN peaks (i.e. MITF peaks), those which showed increased expression upon MITF loss were enriched for aminoglycan, ECM, and axogenesis pathways, whereas those with reduced expression upon MITF loss were enriched for genes involved in pigmentation (*Figure 2d*). We found that 695 of the 1284 induced genes (p<7.3e-09 hypergeometric test) and 535 of the 852 repressed genes (p<6.6e-23, hypergeometric test) associated with MITF peaks (*Figure 2e*; *Supplementary file 2*). Of the 183 ECM and focal adhesion genes whose expression was increased upon MITF knockout, 101 were associated with MITF peaks and induced in expression upon loss of MITF (*Supplementary file 2*). We compared our MITF CUT and RUN peaks with the published MITF ChIP-seq data from COLO829 (generated using the same antibody as used here) (*Supplementary file 3*) and HA-MITF ChIP-seq 501Mel cells (*Supplementary file 4*) and found 42 ECM genes consistently bound by MITF in all three studies (*Figure 2f,g*; *Supplementary file 5*).

To determine if the MITF peaks near induced and reduced genes contained the canonical MITF-binding sites 5'-TCACGTG-3' or 5'-TCATGTGA-3', we performed de novo motif analysis of MITF-bound regions near DEGs using the MEMEChIP tool (*Ma et al., 2014*). We found that MITF peaks

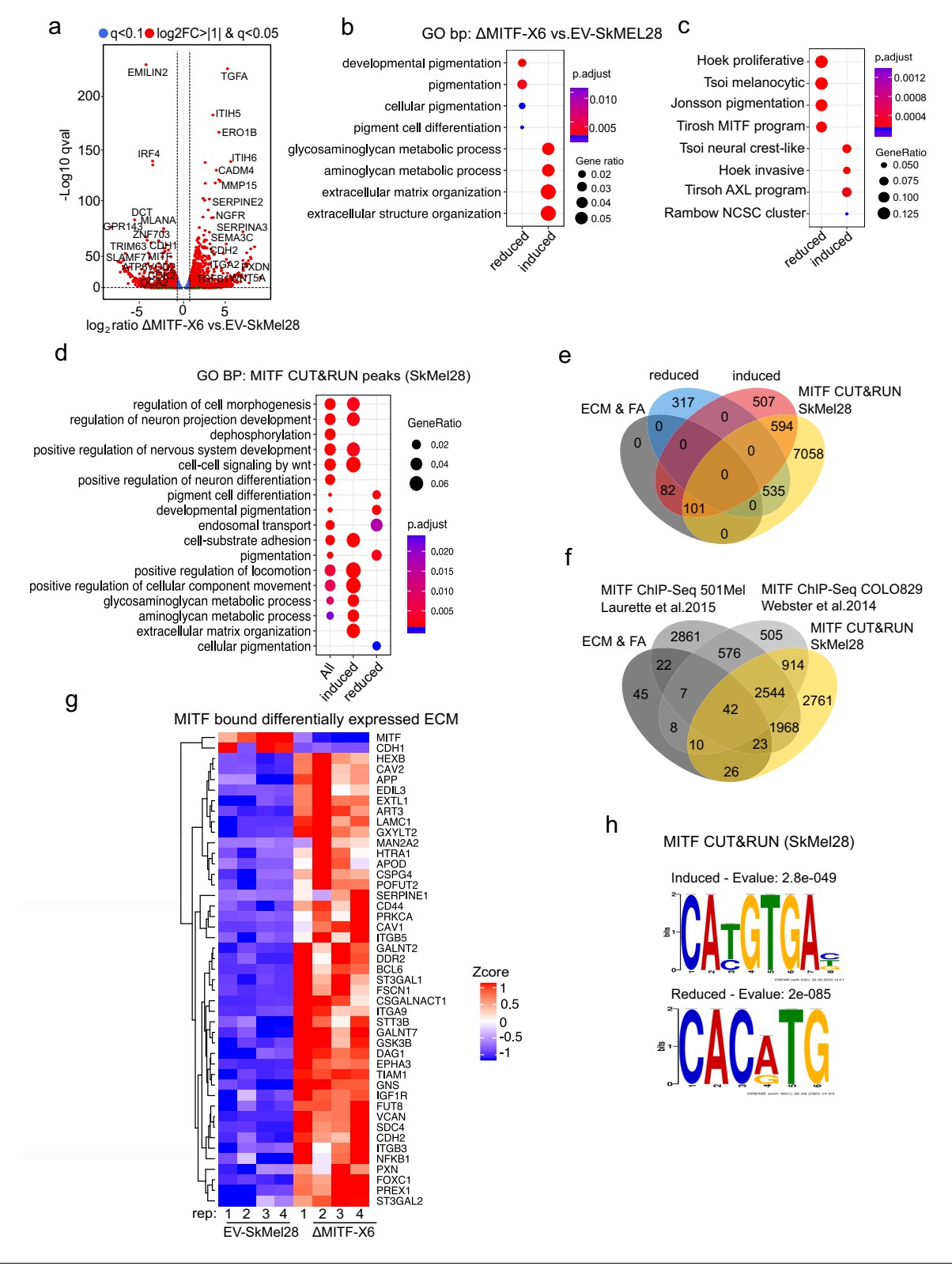

**Figure 2.** MITF binds and represses genes of extracellular matrix (ECM) and focal adhesion genes. (**a**) Volcano plot showing 2136 DEGs with qval <0.5 among which 1516 genes with log2FC≥|1| fold change in expression ΔMITF-X6 vs. EV-SkMel28. (**b**) GO BP analysis of the 1284 induced and 852 reduced DEGs between ΔMITF-X6 vs. EV-SkMel28 cells presented in dot plot; adjusted p-value is red lowest to blue highest; gene ratio is the ratio between DEGs and all genes in the GO category. (**c**) Dot plot of enrichment analysis showing the enrichment of gene signatures from the literature in

*Figure 2 continued on next page*

Figure 2 continued

reduced and induced DEGs of ΔMITF-X6 vs. EV-SkMel28. p Value is red lowest to blue highest; gene ratio is the ratio between genes and all genes in the GO category; reduced: genes reduced in expression in ΔMITF-X6 compared to EV-SkMel28; induced: genes induced in expression in the ΔMITF-X6 compared to EV-SkMel28. (d) GO BP analysis of MITF CUT and RUN peaks associated genes were plotted using Clusterprofiler (*Yu et al., 2012*) in R; All: MITF CUT and RUN peak-associated genes, induced and reduced: Induced or reduced DEGs of ΔMITF-X6 vs. EV-SkMel28 cells based on MITF CUT and RUN peak presence on their gene promoter or distal region binding. (e) Venn diagram showing the overlap between MITF targets identified from MITF CUT and RUN with induced, reduced, ECM and focal adhesion DEGs of ΔMITF-X6 vs. EV-SkMel28 cells. (f) Venn diagram displaying the common overlap between MITF ChIP-seq targets in different cell lines and differentially expressed ECM and focal adhesion genes in ΔMITF-X6 vs. EV-SkMel28 cells. (g) Heatmap showing the differentially expressed ECM genes in ΔMITF-X6 vs. EV-SkMel28 cells that are commonly bound by MITF across different MITF CUT and RUN data sets. Zcore converted TPM value from RNA-seq data was used for plotting. (h) Motif analysis of MITF CUT-and-RUN targets of induced and reduced genes in ΔMITF-X6 vs. EV-SkMel28 cells.

The online version of this article includes the following figure supplement(s) for figure 2:

**Figure supplement 1.** MITF CUT&RUN peak distribution and motifs.

**Figure supplement 2.** MITF directly regulates NGFR and MLANA expression.

associated with induced genes (777) were primarily enriched for the 5'-CA[T/C]GTGAC-3' motif, whereas those near reduced genes (535) were enriched for the 5'-CACATG-3' motif (*Figure 2h*). Thus, genes that are both induced and reduced in expression upon MITF loss contain MITF-binding sites and are likely to be direct targets of MITF. Along with primary motifs, we observed secondary motifs for RUNX1and SOX10 in MITF peaks near induced genes (*Figure 2—figure supplement 1b*), and FOXC1-like motifs in MITF peaks near reduced genes (*Figure 2—figure supplement 1c*). The differences observed in the secondary motifs may represent factors involved in repression versus activation functions of MITF. Taken together, we show that loss of MITF mostly elevates the expression of ECM and focal adhesion genes and a large subset of them are directly bound by MITF.

## MITF depletion leads to increased expression of ECM genes

In order to verify that the link between MITF and the ECM and focal adhesion genes is not restricted to a particular cell line, we performed knockdown and overexpression studies in independent human melanoma cell lines and characterized gene expression data in the Cancer Genome Atlas. First, we performed mRNA sequencing after transient knockdown of MITF in SkMel28 and 501Mel cells, both of which express MITF endogenously at high levels. We identified 1040 DEGs (q-val <0.05, log2FC≥|1|, 567 induced, 473 reduced) upon siRNA-mediated MITF depletion in SkMel28 cells compared to siCTRL and 1114 DEGs in 501Mel cells (q-val <0.05, log2FC≥|1|, 624 induced, 490 reduced) (*Supplementary file 1*). A significant correlation was observed between the DEGs of ΔMITF-X6 vs. EV-Skmel28 cells and DEGs of siMITF vs. siCTRL in SkMel28 (Pearson correlation = 0.66, p<2.2e-16) and 501Mel cells (Pearson correlation = 0.57, p<2.2e-16) (*Figure 3a,b*). Second, we used the Cancer Genome Atlas dataset to characterize differential gene expression and split the tumors into two groups: the tumors with the 10% highest (MITF^high) and 10% lowest (MITF^low) expression of MITF. By performing differential gene expression analysis between the two groups, we identified 2655 DEGs (FDR < 0.01, log2FC≥|1|, 1835 induced and 820 reduced) between MITF^low and MITF^high tumors (*Supplementary file 1*). Interestingly, the DEGs observed when comparing the ΔMITF-X6 cells to the EV-SkMel28 cells and the DEGs observed upon comparing the MITF^low and MITF^high tumors were significantly correlated (R = 0.76, p<2.2e-16) (*Figure 3c*). Additionally, principal component analysis using the top 200 most statistically significant genes in each case revealed that MITF^low tumors cluster near the ΔMITF-X6 cells, whereas MITF^high tumors cluster with EV-SkMEL28 cells, indicating that ΔMITF-X6 cells portray the transcriptional state of MITF^low tumors (*Figure 3e*). Third, we investigated whether overexpression of MITF would lead to repression of ECM genes. To do this, we performed mRNA-sequencing in A375P cells overexpressing a dox-inducible FLAG-tagged MITF construct (pB-MITF-FLAG). A control A375P cell line was generated using an empty vector only expressing FLAG (pB-FLAG). We identified 8110 DEGs (qval <0.05, log2FC≥|1|, 4863 induced, 3247 reduced) between pB-MITF-FLAG and pB-FLAG in A375P cells and among genes that are decreased in expression are ECM-related genes (*Supplementary file 1*). As expected, the DEGs observed upon MITF overexpression in A375P cells showed anti-correlation with the DEGs observed when comparing ΔMITF-X6 to EV-SkMel28 cells (Pearson correlation = −0.46, p<2.2e-16) (*Figure 3d*).

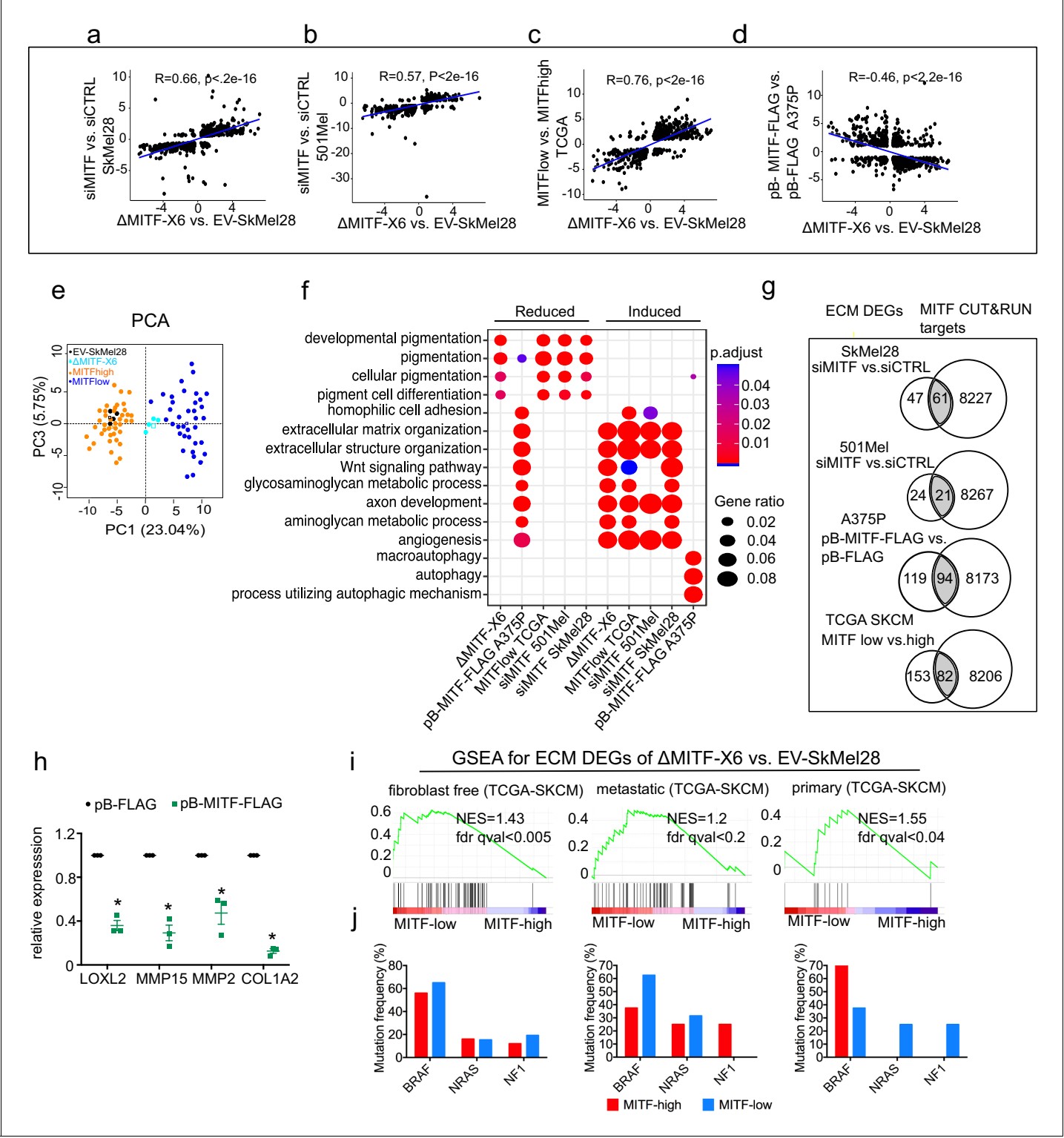

**Figure 3.** The extracellular matrix (ECM) and focal adhesion gene signature is overrepresented upon MITF depletion and in MITF[low] human melanoma tumors. (a–d) Positive correlation of DEGs in ΔMITF-X6 vs. EV-SkMel28 cells with DEGs of siMITF vs. siCTRL in 501Mel and SkMel28 and MITF[low] vs. MITF[high] melanoma tumors from TCGA, and negative correlation of DEGs in pB-FLAG vs. pB-MITF-FLAG A375P cells is shown. Values used in the X and Y axis are log2 fold change in the expression of DEGs. (e) Principal component analysis (PCA) of the 200 most significant DEGs in the MITF[low] vs. MITF[high] and ΔMITF-X6 vs. EV-SkMel28 display similar clustering where EV-SkMel28 samples cluster with MITF[high] tumors and ΔMITF-X6 cells with MITF[low] tumors. (f) GO BP analysis of induced and reduced DEGs affected by MITF KO/KD in SkMel28 and 501Mel cells, and DEGs affected by MITF

*Figure 3 continued on next page*

**Figure 3 continued**

overexpression in A375P cells. (g) Venn diagram displaying the overlap in the number of differentially expressed ECM genes affected by MITF and MITF CUT-and-RUN targets. (h) RT-qPCR showing reduced expression of ECM genes in the stable dox-inducible MITF overexpression A375P cell line (pB-MITF-FLAG). Relative expression was calculated by normalizing to control cells expressing empty vector (pB-FLAG). Error bars indicate standard error of the mean (* p value <0.05) was calculated using paired t-test. (i) Gene set enrichment analysis using ECM genes differentially expressed between ΔMITF-X6 and EV-SkMel28 cells in the top 30 MITF$^{low}$ and 30 MITF$^{high}$ samples with high fibroblast marker removed Primary, and metastatic melanoma from TCGA were analyzed separately. (j) Percentage of mutations in the indicated genes in the MITF$^{low}$ and MITF$^{high}$ tumors from fibroblast-free, primary and metastatic TCGA tumor samples, respectively.

The online version of this article includes the following source data for figure 3:

**Source data 1.** ECM gene expression quantified by qPCR in A375P cells.

To classify genes that are overrepresented after the loss or gain of MITF, we performed GO term enrichment analysis on the DEGs, which revealed an induction of ECM-related genes upon MITF depletion in 501Mel and SkMel28 cells as well as in MITF$^{low}$ tumors, whereas genes involved in pigmentation were reduced in expression (*Figure 3f*). Conversely, overexpressing MITF in the A375P cell line led to a reduction in expression of ECM genes and induction of pigmentation and autophagy genes, again showing that MITF negatively regulates the expression of ECM genes (*Figure 3f*).

Analysis of the MITF ChIP-seq data (*Laurette et al., 2015*) showed that a significant portion of the differentially expressed ECM genes upon MITF KD and in MITF$^{low}$ tumors have MITF peaks in their regulatory domains (*Supplementary file 5*; *Figure 3g*). In contrast, overexpression of MITF led to the repression of 213 ECM genes, 82 of which were direct MITF targets, indicating a major repressive influence of MITF on ECM gene expression (*Figure 3g*; *Supplementary file 5*). We confirmed the repressive effects of MITF by RT-qPCR in dox-inducible A375P cells overexpressing pB-MITF-FLAG, which showed a significant reduction in the expression of *LOXL2*, *MMP15*, *MMP2*, and *COL1A2* when compared to control pB-FLAG cells (*Figure 3h*). Together, our data support our conclusion that MITF is an important direct repressor of ECM gene expression in human melanoma cells and tissues.

Next, we analyzed whether the collagens that were differentially expressed in the MITF-KD or KO melanoma cell lines were also affected by MITF in melanoma tumors in TCGA. Interestingly, using GSEA analysis, we observed increased enrichment of ECM genes in the MITF$^{low}$ tumors (*Figure 3i*). However, to rule out the possibility that the increased expression of ECM genes in TCGA MITF$^{low}$ melanoma tumors was derived from fibroblast cells, we removed the 130 melanoma TCGA samples that showed the highest expression of the fibroblast markers *PDGFRB* and *ACTA2* and then assessed the expression of collagens across the 30 MITF highest and lowest melanoma samples, which consistently showed that expression of genes encoding ECM proteins are strongly enriched in MITF$^{low}$ tumors (*Figure 3i*). The enrichment for ECM genes was observed in both primary and metastatic tumors (*Figure 3i*). We did not observe a correlation between MITF expression and the most common *BRAF*, *NRAS,* and *NF1* mutations found in melanoma (*Figure 3j*), indicating that the gene expression changes observed are controlled via transcriptional regulation, directly or indirectly imposed by MITF. We conclude that reduced MITF expression leads to activation of expression of genes involved in ECM and focal adhesion in melanoma cells and tumors and that in many cases this is through direct binding of MITF to their regulatory regions.

## EMT genes are directly regulated by MITF

Genes involved in the epithelial-to-mesenchymal transition (EMT) process have been shown to play a role in melanoma drug resistance and have been linked to low MITF expression (*Denecker et al., 2014*; *Caramel et al., 2013*). Many EMT-inducing transcription factors, including SNAI2 and ZEB1, repress *CDH1* (E-cadherin) (*Moreno-Bueno et al., 2008*). We tested whether melanoma tumors in TCGA that have high MITF expression also express important EMT regulators. While *SNAI2* was shown to be positively correlated with MITF, the expression of *CDH1* did not correlate with the expression of *SNAI2* in the TCGA melanoma tumors which is consistent with the published findings in melanoma and melanocyte samples (*Shirley et al., 2012*), despite *CDH1* being a canonical target of *SNAI2* (*Cano et al., 2000*). Thus, E-cadherin is likely directly regulated by MITF and ZEB1. As expected the analysis of the TCGA data showed that the expression of *CDH2* (*N-cadherin*) (R = −0.3, p<1.9e-11), *TGFB1*(R = −0.49, p<2.2e-16) and *ZEB1* (R = −0.41, p<2.2e-16) was anti-

correlated with MITF in melanoma tumors, whereas the expression of *CDH1* (R = 0.42, p<2.2e-16) and *SLUG* (*SNAI2*) (R = 0.43, p<2.2e-16) was positively correlated (*Figure 4a*). Consistent with this, the expression of the *CDH1* and *SNAI2* genes was reduced in MITF[low] tumors and ΔMITF-X6 cells, whereas the expression of *CDH2*, *SOX2*, *TGFß1*, and *ZEB1* was increased (*Figure 4b*). We also observed increased expression of *CDH2* upon siRNA-mediated KD of MITF in SkMel28 and 501Mel cells; however, the level of *CDH1* was decreased only in the siMITF SkMel28 cells (*Figure 4b*). Interestingly, upon MITF overexpression in the pB-MITF-FLAG A375P cells, the expression of *CDH2*, *SNAI2*, *SOX2*, and *TGFß1* was decreased, whereas the expression of *CDH1* and *ZEB1* was increased (*Figure 4b*). RT-qPCR analysis of EMT genes in the MITF-KO cells confirmed that *CDH1* expression was reduced 50- and 100-fold in the ΔMITF-X2 and ΔMITF-X6 cells, respectively, whereas *CDH2* and *TGFß1* were significantly increased when compared to EV-SkMel28 cells (*Figure 4c*). Western blot analysis confirmed increased expression of the classical EMT marker protein CDH2 and decreased expression of CDH1 in both MITF-KO cell lines (*Figure 4d,e*). Analysis of CUT and RUN and publicly available MITF ChIP-seq data showed that *ZEB1*, *SOX2*, *CDH1*, and *CDH2* genes contain MITF-binding peaks in their intronic and promoter regions (*Figure 2—figure supplement 1d*; *Laurette et al., 2015*), whereas *TGFß1* does not. This suggests that MITF is not only involved in regulating the expression of ECM genes but may also be directly involved in regulating the expression of EMT genes, resulting in EMT-like changes in cell morphology and behavior.

## MITF-mediated effects on ECM genes are reversible

The MITF rheostat model predicts that different levels of MITF activity modulate distinct phenotypic states of melanoma cells and that these effects are reversible (*Lister et al., 2014*). To determine if the effects of long-term MITF knockout could be reversed, we performed a rescue experiment by introducing an exogenous MITF-FLAG or EV-FLAG construct into the MITF-KO cells and then used RT-qPCR to characterize the expression pattern of ECM genes. As expected, the control EV-FLAG transfected MITF-KO cells exhibited increased expression of the ECM genes *CDH2*, *ID1*, and *MMP15* as compared to the EV-SkMel28 control cells (*Figure 5a–d*), whereas the expression of *CDH1* was reduced. Importantly, the expression of all four genes was partially rescued upon introducing the MITF-FLAG construct into ΔMITF-X6 cells; a smaller rescue effect was observed in ΔMITF-X2 cells transfected with MITF-FLAG (*Figure 5a–d*).

In order to overcome the partial rescue seen with the MITF-KO cells, we used the piggybac transposon system to integrate a dox-inducible synthetic micro-RNA construct (*miR-MITF*) into 501Mel and SkMel28 cells, thus allowing inducible knockdown (KD) of MITF by addition of doxycycline (*Figure 5e*). At the same time, cells carrying a non-targeting control (*miR-NTC*) were generated. We induced MITF-KD in the miR-MITF SkMel28 cell line by adding dox and removed it again after 24 hr to assay for gene and protein expression at defined time points (*Figure 5e*). We chose to focus on the ECM and EMT genes *CDH1*, *CDH2*, *ITGA2*, and *SERPINA3*, all of which are direct targets of MITF (*Laurette et al., 2015*). Our results showed that MITF mRNA and protein expression was significantly decreased after 24 hr of dox treatment and reached basal levels again 96 hr after dox removal (*Figure 5f,g*), showing that the dox-inducible system is suitable for reversibly modulating MITF levels. We observed a sharp decrease in *CDH1* mRNA expression after 24 hr of dox treatment. However, 72 and 96 hr after dox removal its expression had gradually increased, consistent with the restoration of *MITF* expression (*Figure 5g*). Similarly, the expression of genes repressed by MITF such as *CDH2*, *SERPINA3*, and *ITGA2* was sharply increased after 24 hr of dox treatment and decreased again 96 hr after dox removal (*Figure 5h–j*). Western blotting showed that the expression of the E-cadherin (CDH1) protein was reduced, whereas the expression of N-Cadherin (CDH2) was increased when compared to the *miR-NTC* control (*Figure 5f*). After 72 and 96 hr of dox removal, MITF expression was restored and expression of the E-Cadherin protein was increased back to initial levels, whereas the expression of N-Cadherin was reduced compared to that observed at 24 hr of MITF KD (*Figure 5f*). These data show that, consistent with the rheostat model, the function of MITF as both a repressor and activator of gene expression has reversible effects on the expression of EMT and ECM genes.

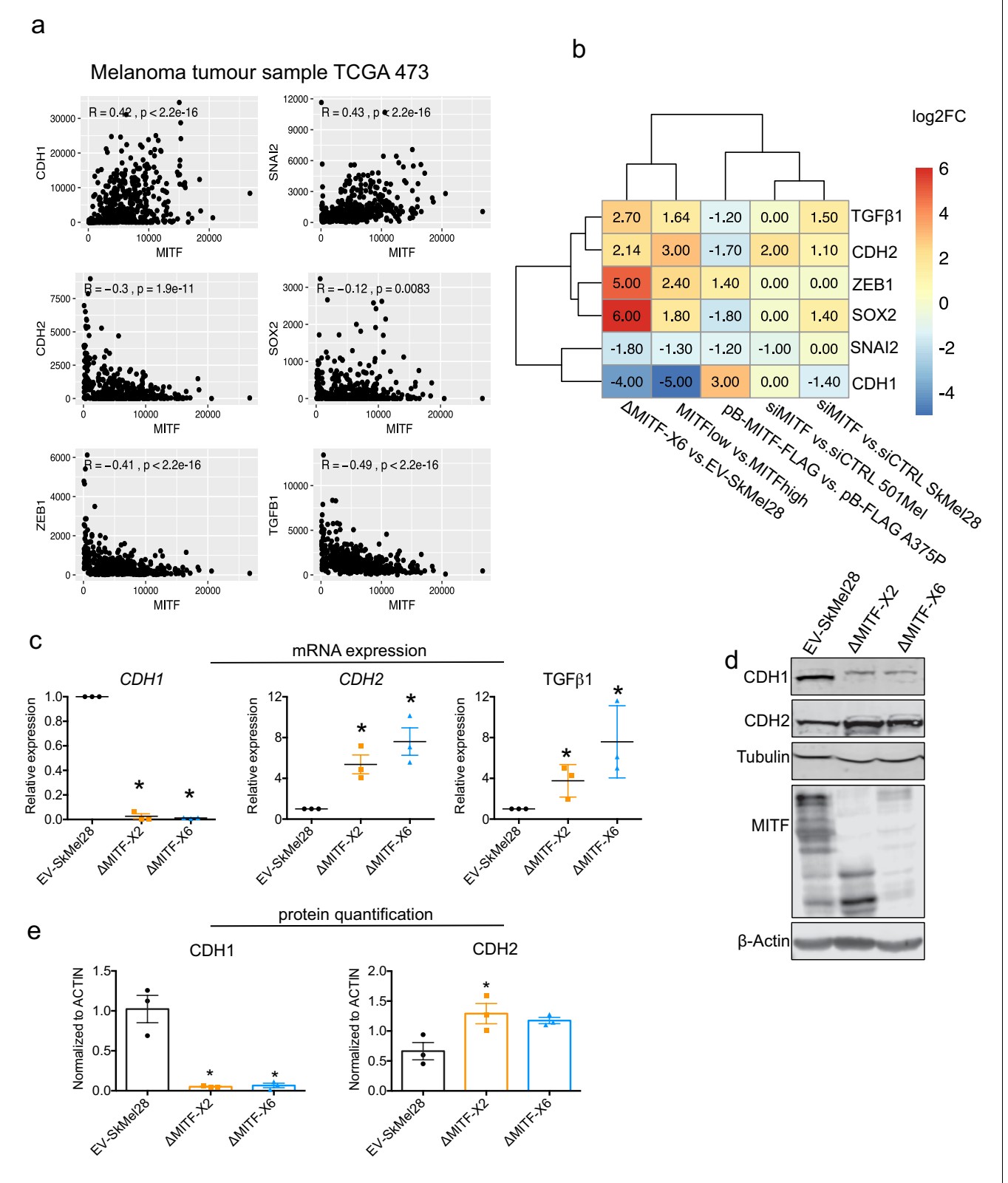

**Figure 4.** Epithelial-to-mesenchymal transition (EMT) genes are directly regulated by MITF. (**a**) Scatter plot displaying the Spearman correlation between MITF mRNA expression with EMT genes in the 472 melanoma tumor samples from TCGA; MITF displayed positive correlation with *CDH1* and *SNAI2* and negative correlation with *ZEB1*, *TGFβ1*, and *CDH2*. (**b**) Differentially expressed EMT genes plotted as heatmap using the log2 fold change value of DEGs of MITF depletion in SkMel28 and 501Mel cells, MITF overexpression in A375P cells and MITF[low&high] melanoma tumors. (**c**) Real-time

*Figure 4 continued on next page*

*Figure 4 continued*

qPCR (RT-qPCR) evaluation of EMT genes in the EV-SkMel28, ΔMITF-X2, and ΔMITF-X6 cell lines. Fold change in the expression calculated over EV-SkMel28. Error bar represents standard error of the mean (* p value <0.05) was calculated using one-way ANOVA (multiple correction with Dunnett test). (d, e) Western blot analysis and quantification (Fiji Image J) of protein expression of CDH1, CDH2, and MITF in EV-SkMel28, ΔMITF-X2, and ΔMITF-X6 cell lines. ß-Actin was used as loading control. * p value <0.05 was calculated by one-way ANOVA (multiple correction with Dunnett test). The online version of this article includes the following source data for figure 4:

**Source data 1.** Quantification of CDH1 and CDH2 protein and mRNA in MITF-KO cells.

## MITF affects the number of focal adhesions

Based on the observed increase in the expression of ECM and focal adhesion genes, we expected focal adhesion formation to be affected in the MITF-depleted cells. Indeed, immunostaining revealed an increased number of paxillin (PXN)-positive focal points (stained using PXN phospho - Tyr118 antibodies) around the cell periphery of MITF-KO cells as compared to EV-SkMel28 control cells (*Figure 6—figure supplement 1a*). Quantification of the focal points showed around twofold increase in their numbers in both MITF-KO cell lines (*Figure 6—figure supplement 1b*). Transcriptomic data of the 473 melanoma tumor samples from TCGA showed a significant negative correlation between the expression of *MITF* and *PXN* in these samples (*Figure 6—figure supplement 1c*). We also assessed the expression of *PXN* in a panel of 163 patient-derived melanoma cells exhibiting different levels of *MITF*. This showed that expression of *PXN* was specifically induced in MITF$^{low}$ melanoma cell lines and displayed a negative correlation with *MITF* expression (*Figure 6—figure supplement 1d,e*). In order to evaluate whether the formation of focal adhesions would be induced upon short-term MITF loss, we integrated the dox-inducible *miR-MITF* transgene into 501Mel and SkMel28 cells and detected focal adhesions using the PXN antibody. After a 24 hr induction of MITF KD, a twofold increase was observed in the number of PXN-positive focal points at the cell borders when compared to the *miR-NTC* control cell lines (*Figure 6—figure supplement 1f–h*). Analysis of ChIP-seq data showed an MITF peak in intron 6 of *PXN* containing the CACGTG motif (*Figure 6—figure supplement 1i*). This indicates that MITF affects the formation of focal adhesions by directly regulating the expression of *PXN*, a key player in focal adhesion.

Previous studies have shown that adaptive resistance to the BRAF$^{V600E}$ inhibitor vemurafenib leads to activation of focal adhesion and ECM-related pathways (*Fallahi-Sichani et al., 2017*). Indeed, treating the cells with vemurafenib led to a decrease in MITF protein expression in EV-SkMel28 cells which is consistent with the literature. However, the expression of MITF in the 501Mel cell upon vemurafenib treatment was increased compared to a DMSO control (*Figure 6—figure supplement 2a,b*). This raises the question of whether the effects observed on ECM and focal adhesion genes upon BRAF inhibition are mediated through MITF. To evaluate the effects of BRAF inhibition on focal adhesions, we treated MITF-KO and EV-SkMel28 cells with vemurafenib and stained for phospho-paxillin (Tyr118). Consistent with the observation above, the MITF-KO cells showed a fourfold increase in the number of focal adhesions as compared to EV-SkMel28 cells under the control DMSO-treated conditions (*Figure 6a* (upper panel), b). Treatment with vemurafenib resulted in a significant increase in the number of focal adhesions in the EV-SkMel28 cells but a further increase was also observed in the MITF-KO cells (*Figure 6a* [lower panel], b). Consistent with this, knockdown of MITF induced through the miR-MITF construct in both 501Mel and SkMel28 cells led to an increased number of focal adhesions when compared to *miR-NTC* cells (*Figure 6c,d* [upper panels], e, f). Treatment with vemurafenib further increased the number of focal adhesions in SkMel28 cells expressing miR-NTC or miR-MITF, but again, more focal points were observed in miR-MITF cells under these conditions (*Figure 6c,d* [lower panels]). Importantly, vemurafenib treatment alone did not lead to an increase in focal adhesion formation in 501Mel cells expressing *miR-NTC* which is consistent with increased MITF protein expression upon vemurafenib treatment, whereas a major increase in focal adhesions was observed upon MITF depletion in the miR-MITF cells (*Figure 6c* [lower panel], e). These results suggest that the formation of focal adhesions upon vemurafenib treatment is in part dependent on changes in MITF expression. However, since a further increase is observed in upon vemurafenib treatment of the knockout and knockdown lines, other factors must also be involved.

Since an increase in formation of focal adhesions is a marker for MITF loss, we sought to inhibit formation of focal adhesion using increasing concentration of FAK inhibitor in MITF-KO and EV-

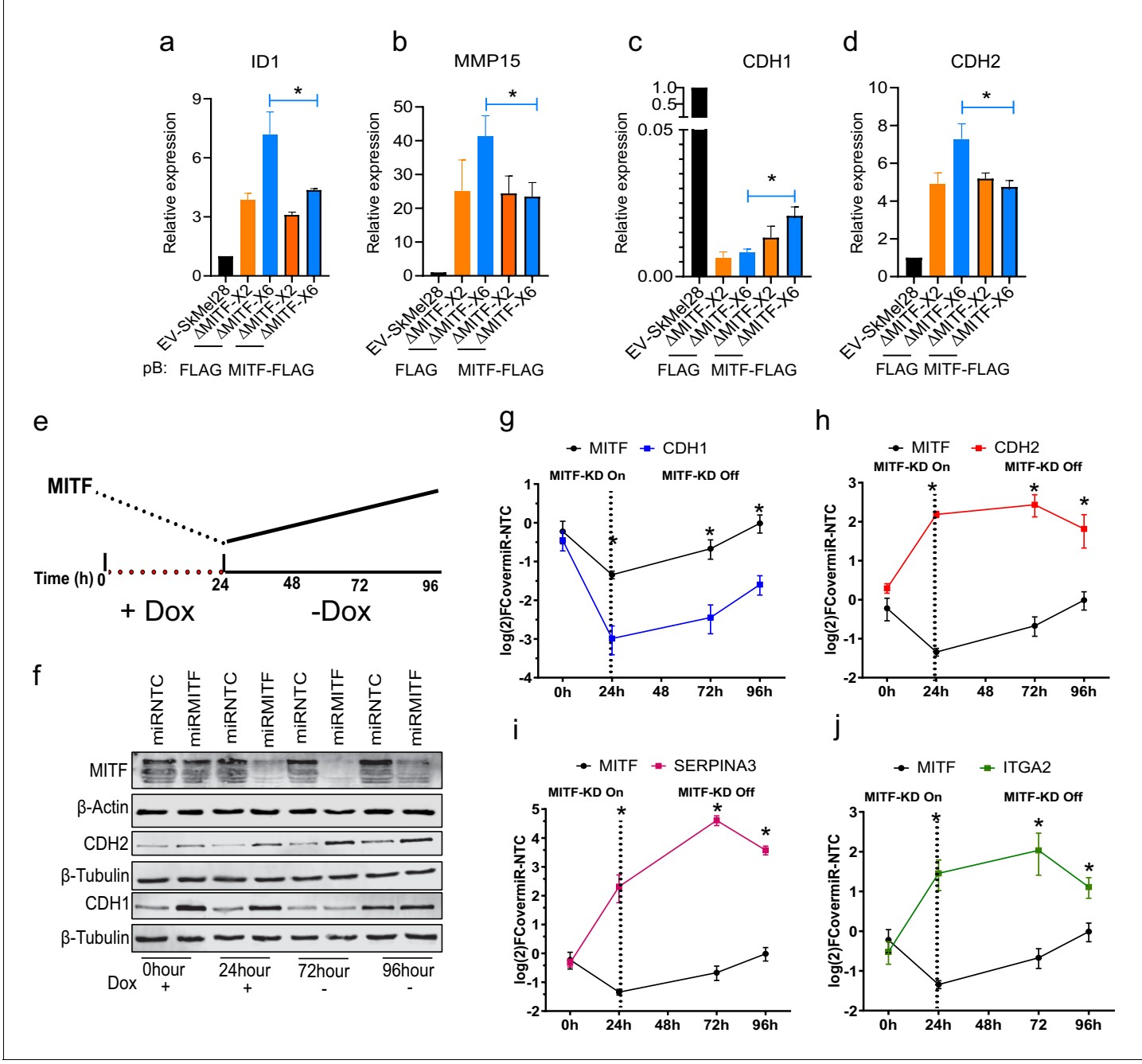

**Figure 5.** The effects of MITF on epithelial-to-mesenchymal transition (EMT) and extracellular matrix (ECM) gene expression are reversible. (a–d) Gene expression of ECM and EMT genes evaluated by RT-qPCR in EV-SkMel28 and MITF-KO cells with ectopic expression of EV-FLAG and MITF-FLAG constructs. Expression was normalized to EV-SkMel28 cells. Error bars represent standard error of the mean, * p value <0.05 was calculated by two-way ANOVA (multiple correction with Sidak test). (e) Schematic showing the dox-inducible MITF KD system. MITF expression decreases in the presence of dox (first 24 hr) and reverts back to baseline levels upon dox wash-off (at 72–96 hr). (f) Western blot analysis for the protein expression of MITF and CDH1, and CDH2 with the presence of dox treatment 0 and 24 hr or absence of dox 72 and 96 hr. (g–j) RT-qPCR analysis of MITF targeted ECM genes in miR-NTC and miR-MITF SkMel28 cells, treated with dox for 24 hr to induce MITF KD and after dox wash-off at 72 and 96 hr. Expression was normalised to miR-NTC cell lines, error bars represent standard error of the mean, * p value <0.05 was calculated by two-way ANOVA (multiple correction with Sidak test).

The online version of this article includes the following source data for figure 5:

**Source data 1.** Time profile of ECM gene expression upon MITF KD.

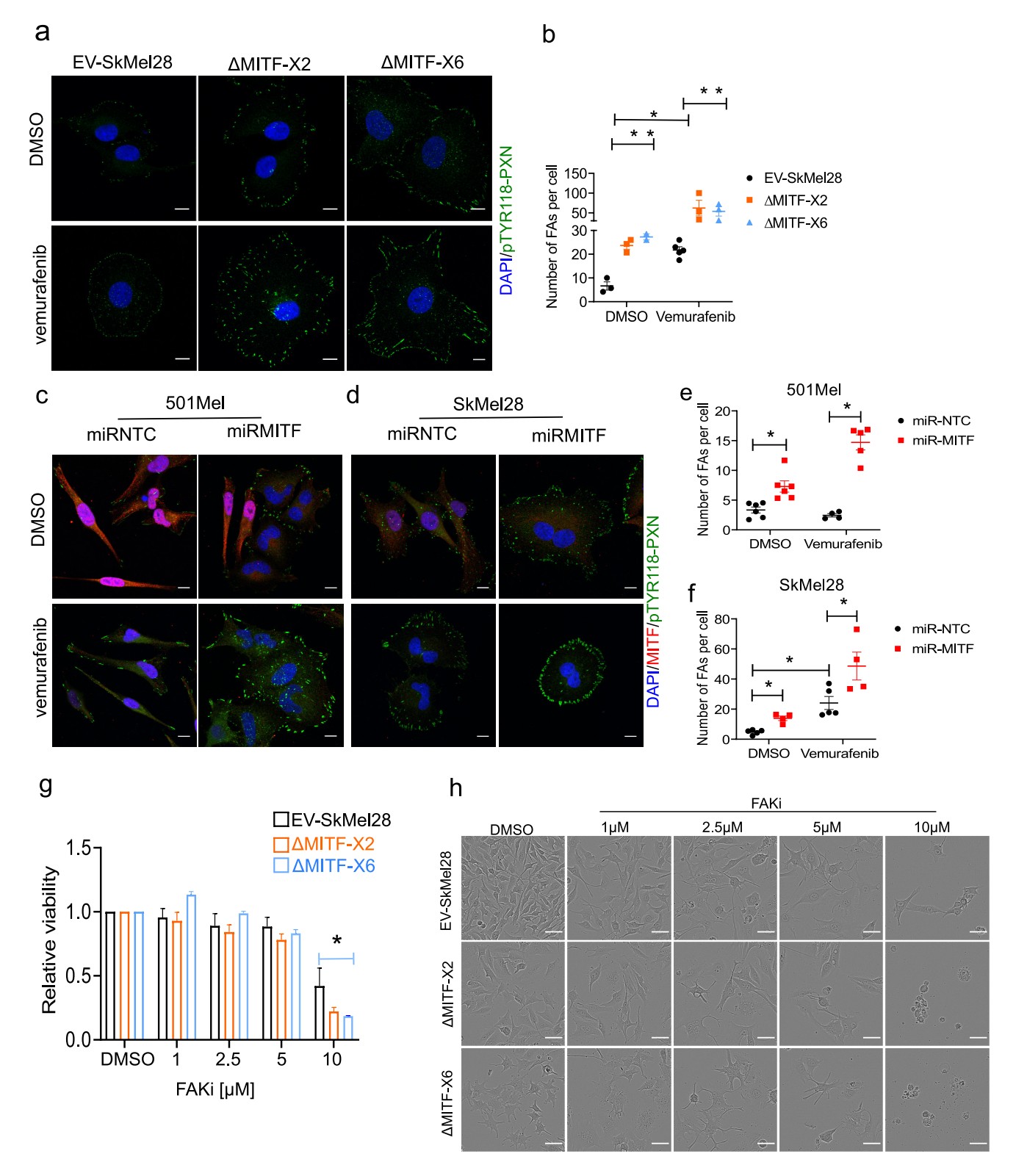

**Figure 6.** MITF mediates formation of focal adhesion. (a–b) Immunostaining for p-PXN$^{TYR118}$ and quantification of p-PXN$^{TYR118}$-positive focal points in EV-SkMel28, ΔMITF-X2 and ΔMITF-X6 cell lines treated with DMSO (a, upper panel) or vemurafenib (a, lower panel). (c–d) Immunostaining for p-PXN$^{TYR118}$ and MITF and (e, f) quantification of p-PXN$^{TYR118}$-positive focal points in miR-NTC and miR-MITF treated 501Mel and SkMel28 cells. Error bars represent standard error of the mean, * p value <0.05 was calculated by two-way ANOVA (multiple correction with Sidak test). (g) Bar plot

*Figure 6 continued on next page*

*Figure 6 continued*

represents the relative viable cells compared to DMSO control with increasing concentration of FAK inhibitor (PF562271) for 72 hr. Error bars represent standard error of the mean, * p value <0.05 was calculated by two-way ANOVA (multiple correction with Tukey test). (**h**) Incucyte images of cells after 72 hr treatment with increasing concentrations of FAKi (PF562271) and with respective DMSO control.

The online version of this article includes the following source data and figure supplement(s) for figure 6:

**Source data 1.** Quantification of focal adhesions upon vemurafenib treatment.

**Figure supplement 1.** MITF represses PXN in melanoma cell lines.

**Figure supplement 1—source data 1.** Number of p-PXN (TYR118) positive focal points in MITF-KO and KD cells.

**Figure supplement 2.** MITF-KO cells mimic MRD in melanoma.

**Figure supplement 2—source data 1.** Westernblot quantification upon vemurafenib treatment.

SkMel28 cells. We observed that after 72 hr of treatment with 10 µM of FAKi the viability of MITF-KO cells was significantly reduced compared to the EV-SkMel28 cells; the effect of FAKi on ∆MITF-X6 cells was more pronounced than in ∆MITF-X2 cells (*Figure 6g,h*). This suggests that the survival of MITF-KO cells is dependent on FAK signaling.

To understand whether the ECM and focal adhesion genes affected upon MITF loss overlap with the gene signature of melanoma cells that have been treated with BRAF inhibitors, we used single-cell RNA-sequencing data of human melanoma xenografts (*Rambow et al., 2018*). We focused on gene signatures specific for single-cell populations with low MITF, (i) a subpopulation of cells which represent minimal residual disease (MRD) in melanoma, a small population of cells that remain upon drug treatment and (ii) an invasive gene signature (*Rambow et al., 2018*). Our GSEA analysis showed that ∆MITF-X6 cells were significantly enriched in the MRD gene signature but not with the invasive signature found in another sub-population of MRD cells in the xenografts (*Figure 6—figure supplement 2c*). Among the genes that overlap between the MRD and ∆MITF-X6 cells are ECM genes such as *COL4A1*, *ITGA1*, *ITGA6*, *LAMC1*, and *VCAN*. The same findings were obtained using single cell RNA-seq data of MITF-depleted zebrafish melanomas as well as bulk-RNA-seq data of MITF$^{low}$ melanoma tumors (*Travnickova et al., 2019*). Both datasets showed positive enrichment with ∆MITF-X6 cells (*Figure 6—figure supplement 2d*). Importantly, we found that in the zebrafish data the ECM signature was specifically induced in the single cell cluster from MITF-low superficial tumors (representing minimal residual disease) compared to other single-cell clusters from MITF-high melanomas (*Figure 6—figure supplement 2e*). These results suggest that the loss of MITF is an important mediator of MRD in melanoma and that MRD cells alter their extracellular environment.

## MITF KO affects proliferation and migration

The rheostat model predicts that MITF loss should reduce cell proliferation but increase migration potential of melanoma cells. We therefore measured proliferative ability of the MITF-KO cells using different methods. First, we characterized cell confluency over time using IncuCyte live cell imaging. This showed that both of the MITF-KO cells had a twofold reduction in proliferation rate as compared to the EV-SkMel28 cells (*Figure 7a*). Second, a BrdU incorporation assay showed that ∆MITF-X6 and ∆MITF-X2 cells had fewer (20–25%) BrdU-positive cells than the EV-SkMel28 cells (45%), suggesting that there are fewer actively proliferating cells in the MITF-KO cells compared to the control cells (*Figure 7b*).

Previous analysis has shown that knocking down MITF leads to increased migration ability of melanoma cells (*Carreira et al., 2006*; *Giuliano et al., 2010*; *Cheli et al., 2012*; *Javelaud et al., 2011*; *Bianchi-Smiraglia et al., 2017*; *Falletta et al., 2017*). We therefore characterized the migration ability of our knockout cells. Strikingly, the wound scratch assay showed that the MITF-KO cells failed to close the wound in 24 hrs, whereas the EV-SkMel28 cells were able to close the wound within that time (*Figure 7c,d*). To test whether the effects on migration were due to the long-term depletion of MITF in the MITF-KO cells, we performed the wound scratch assay upon MITF KD in the miR-MITF cells. Upon MITF-KD, we observed a minor decrease in the ability of the cells to close the wound when compared to the control miR-NTC cells (*Figure 7e,f*). Next, we assessed the invasion ability of the MITF-KO cells using transwell chambers coated with matrigel. Interestingly, we found that MITF-KO cells displayed a severe reduction in invasion ability compared to the EV-SkMel28 cells

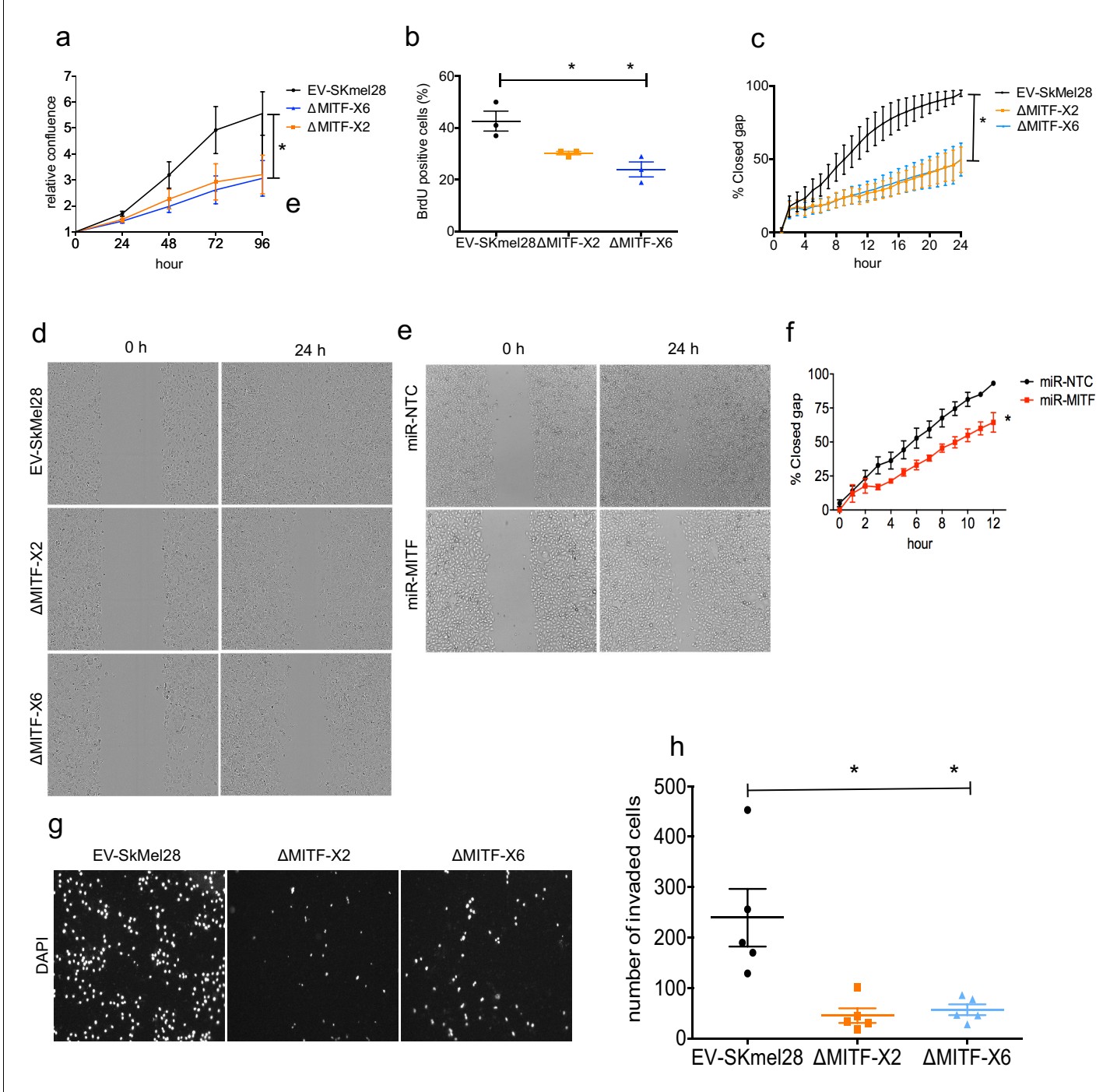

**Figure 7.** MITF knockout affects proliferation, migration, and invasion ability of melanoma cells. (a) Relative cell confluency obtained from IncuCyte live cell imaging compared to day 0 was plotted for EV-SkMel28, ΔMITF-X2, and ΔMITF-X6 cell lines; Error bars represent standard error of the mean, * p value <0.05 was calculated by one-way ANOVA. (b) Percentage of BrdU-positive cells was assessed by flow cytometry in EV-SkMel28, ΔMITF-X2, and ΔMITF-X6 cell lines. Error bars represent standard error of the mean, * p value <0.05 was calculated by one-way ANOVA (multiple correction with Dunnett test). (c, d) Quantification and images of wound scratch assay in EV-SkMel28, ΔMITF-X2, and ΔMITF-X6 cells over 24 hr time period. Error bars represent standard error of the mean, * p value <0.05 was calculated by two-way ANOVA (multiple correction with Sidak test). (e, f) Quantification and images of wound scratch assay in miR-NTC and miR-MITF in SkMel28 cells over 12 hr time period. Error bars represent standard error of the mean, * p value <0.05 was calculated by one-way ANOVA (multiple correction with Sidak test). (g, h) Matrigel invasion assay of EV-SkMel28, ΔMITF-X2, and ΔMITF-X6 cells using the transwell assay; Quantification of invaded cells per transwell filter. Error bars represent standard error of the mean, * p value <0.05 was calculated by one-way ANOVA (multiple correction with Dunnett test).

The online version of this article includes the following source data for figure 7:

**Source data 1.** Quantification of cell proliferation, migration and invasion of MITF-KO and KD cells.

(*Figure 7g,h*). Taken together our data suggests that knocking down MITF negatively influences both cell proliferation and migration ability of the cells.

## Discussion

In this study, we have shown that MITF directly binds to and represses the expression of ECM, EMT and focal adhesion genes in human melanoma cells. We first observed this using our MITF-KO cells but verified our observations in other cell models by overexpression and knockdown of MITF using siRNA and inducible microRNA against MITF (*miR-MITF*) in melanoma cells. Importantly, we showed that MITF^low tumors in humans as well as in zebrafish have increased expression of ECM and focal adhesion genes. Together, our findings indicate that MITF acts as a transcriptional repressor of genes involved in ECM and focal adhesion.

A role for MITF as a repressor has been described in both melanoma cells and immune cells (*Riesenberg et al., 2015*; *Hu et al., 2007*). In myeloid precursor cells, MITF was shown to interact with EOS to recruit co-repressors to target genes (*Hu et al., 2007*), whereas in melanoma cells MITF bound directly to an E-box located in an enhancer of the *c-JUN* gene, leading to reduced expression of the gene (*Riesenberg et al., 2015*). Our results show that many of the genes whose expression is repressed by MITF are bound by MITF and contain E-boxes in their regulatory regions (*Figure 2h*). This suggests that direct binding of MITF is involved in their repression. Since we observed differences in secondary motifs between the repressed and activated genes, different co-factors may be involved in mediating the repression in each case.

The MITF-dependent rheostat model predicts that high MITF activity promotes proliferation, whereas low activity promotes invasion (*Carreira et al., 2006*). Consistent with the rheostat model, proliferation was severely reduced upon MITF knockout (*Figure 7a,b*). Unexpectedly, however, the migrative and invasive properties were reduced in both MITF-KO and MITF-KD (*miR-MITF*) cells (*Figure 7c–h*). Immunohistochemistry and single-cell sequencing studies of melanoma tumors have shown the existence of cells with low or no MITF expression (*Goodall et al., 2008*; *Rambow et al., 2018*; *Travnickova et al., 2019*). The involvement of MITF in migration has mostly been characterized using knockdown studies in melanoma cell lines using either siRNA or shRNA and by using Matrigel-coated Boyden chambers (*Carreira et al., 2006*; *Bianchi-Smiraglia et al., 2017*; *Cheli et al., 2011*); in these studies, knocking down MITF resulted in increased migration properties. *Cheli et al., 2012* also injected melanoma cells into the tail vein of mice and showed increased formation of metastasis when MITF was knocked down. Two different pathways involved in migration were shown to be regulated by MITF; DIAPH1, a gene implicated in actin polymerization (*Carreira et al., 2006*), and the guanosine monophosphate reductase (GMPR) gene encoding an enzyme involved in regulating intracellular GTP levels (*Bianchi-Smiraglia et al., 2017*). Surprisingly, more recent studies by *Falletta et al., 2017* and *Vlčková et al., 2018* failed to observe any effects on migratory/invasive properties upon MITF knockdown using the same cell lines as were used in the previous studies. *Falletta et al., 2017* suggested that the translation factor eIF2α was needed along with the nutrient sensor ATF4 to mediate invasion of melanoma cells (*Falletta et al., 2017*). Clearly, the experimental system used and additional triggers such as nutrient limitations might play a role in mediating the invasive phenotype. Interestingly, knocking down SMAD7 in melanoma cells resulted in a dual invasive-proliferative phenotype without affecting MITF expression (*Tuncer et al., 2019*).

Thus, we might speculate that the migratory phenotype is a transient event and that in order to achieve migration the cells need to be tested at a narrow time interval where MITF activity is decreasing. Carmit Levy's group has recently shown MITF oscillations upon UV exposure in order to synchronize stress response and pigmentation (*Malcov-Brog et al., 2018*). It is possible that such a mechanism exists, for example during melanoblast development where oscillating MITF expression might ensure that proliferation and migration can both take place but not at the same time. Previous work has suggested oscillations in the dependence of melanoblasts on the receptor tyrosine kinase KIT (*Yoshida et al., 1996*; *Hou et al., 2000*). It is plausible that activation of invasion genes requires discontinuous presence of MITF along with other co-factors at specific time windows. According to the phenotype switching model of melanoma the reversible switch between proliferative and invasive state is needed for melanoma progression (*Arozarena and Wellbrock, 2019*). In our case, the complete loss of MITF might trap the cells in a state where the MITF switch is dead and migration

therefore not possible. This would then explain why we do not see the invasive gene signature and no changes in the expression of *DIAPH1*, *p27*, or *GMPR*. Our analysis of published transcriptomic signatures of melanoma cells and tumors validated that upon MITF loss the expression of differentiation and proliferation genes was diminished and the expression of drug resistant and neural crest-like program was enriched (*Figure 2c*). Importantly, comparing our gene signature with that of 86 melanoma cells (*Hoek et al., 2006*) showed that the ΔMITF-X6 cells displayed a more prominent loss of proliferative genes than a gain of invasive genes (*Figure 2c*). Similarly, *Rambow et al., 2015* showed that melanoma cell lines (G1, T1, 501Mel, MNT-1, SKMel3) which express high levels of MITF displayed a high proliferation potential but less migratory, invasive, and subcutaneous tumor growth capabilities than cell lines with low MITF expression (WM1366, WM793, WM852, LU1205, A375M) (*Rambow et al., 2015*). The expression of TRIM63 and CAPN3 was over-represented in the cell lines with high MITF expression and knocking down MITF led to a reduction of their expression. Interestingly, knocking down either TRIM63 or CAPN3 enhanced the invasive potential of 501Mel cells. In the ΔMITF-X6 cells, we observed reduction only of TRIM63 expression (*Rambow et al., 2015*). Thus, the transcriptome of our ΔMITF-X6 cells does not fully recapitulate the gene signature of invasive cells.

We identified MITF as an important transcriptional regulator of ECM and focal adhesion genes. Interestingly, we observed increased expression of TGFß1, encoding an important regulator of ECM-related genes, in the MITF-KO cells and MITF<sup>low</sup> melanoma tumors (*Figure 4b,c*). It has been shown that TGFß1 supresses the expression of MITF in melanoblasts, thereby inhibiting differentiation into melanocytes (*Nishimura et al., 2010*). This autocrine signaling of TGFβ is retained in melanoma cells (*Javelaud et al., 2008*). According to *Hoek et al., 2006*, the MITF<sup>low</sup> transcriptional state is dictated by TGFß1 signaling, which can suppress MITF expression resulting in an invasive and drug-resistant phenotype (*Hoek et al., 2006*; *Miskolczi et al., 2018*). This suggests that the genes induced upon MITF loss are partly due to induction of TGFß signaling. However, our results suggest that MITF is directly involved in mediating the observed effects on the expression of ECM and focal adhesion genes. In addition, the relationship between MITF and the expression of TGFß1 is not clear. Our observations suggest that knocking down MITF leads to a major increase in *TGFß1* mRNA expression in the melanoma cells, suggesting that the effects are cell-autonomous and driven by MITF. However, there are no MITF-peaks in or near the *TGFß1* gene in melanoma cells, leading us to hypothesize that the effects must be mediated through a hitherto unknown intermediary.

Enhanced expression and phosphorylation of paxillin has been linked to therapy resistance in other cancer cell types, such as lung cancer (*Wu et al., 2016*). In melanoma, an inverse relation between BRAF inhibition and the expression of ECM genes has been described as a marker of de-differentiated drug-resistant cells (*Fallahi-Sichani et al., 2017*). Our data showed that the number of paxillin-positive dots was induced in both MITF-KO and miR-MITF cells as compared to controls (*Figure 6—figure supplement 1a,b,f,g* h) and paxillin expression was inversely correlated with MITF expression in melanoma tissues and cell lines (*Figure 6—figure supplement 1c–e*). Interestingly, we found that treating cells devoid of MITF with a BRAF inhibitor resulted in an increase in formation of focal adhesions (*Figure 6a–f*). It is worth mentioning that an increase in the number of focal adhesions was restricted to SkMel28 melanoma cells in which MITF protein level was reduced upon vemurafenib treatment (*Figure 6d,f*, *Figure 6—figure supplement 2a, b*). However, we did not observe a significant increase in the number of focal adhesions in the 501Mel cells that gained MITF upon vemurafenib treatment (*Figure 6c,e*, *Figure 6—figure supplement 2a, b*). This highlights the role of MITF as a mediator of focal adhesion formation. However, how the synergistic effects of MITF and vemurafenib on focal adhesion formation are mediated is unclear. One way to explain an increase in the formation of focal adhesions is that it is due to integrin clustering that is essential for the activation of focal adhesion pathways (*Harburger and Calderwood, 2009*; *Humphries et al., 2006*). We observed an increase in the expression of several integrins including *ITGA1*, *ITGA2*, *ITGA6*, *ITGA10*, and *ITGB3* in the MITF-KO cells, as well as in the siMITF 501Mel and SkMel28 cell lines (*Supplementary file 5*). In addition to this, the FLT1 receptor tyrosine kinase (VEGFR1) and its ligand VEGFA, which activate a pathway that phosphorylates FAK, a key mediator of focal adhesions, were increased in expression. Interestingly, both *FLT1* and *VEGFA* have MITF-binding sites in their promoters and MITF has previously been shown to regulate *VEGFA* expression (*Louphrasitthiphol et al., 2019*). Exposure of melanoma cells to BRAF and MEK inhibitors has been shown to slow growth and result in increased expression of *NGFR* and ECM and focal adhesion

genes (*Fallahi-Sichani et al., 2017*). Consistent with these findings, we observed an up to 200-fold induction of the *NGFR* transcript in the MITF-KO cells compared to EV-SkMel28 cells, and we identified an MITF peak in the 3'UTR of *NGFR* in both the MITF CUT-and-RUN data from SkMel28 cells and in the MITF ChIP-seq data from COLO829 cells (*Webster et al., 2014*); expression of the melanocyte differentiation marker and MITF target *MLANA* was 50- to 80-fold reduced in the MITF-KO cells (*Figure 2—figure supplement 2a–d*). Thus, it is possible that MITF affects focal adhesions by both directly regulating expression of genes involved in the process and indirectly by activating the expression of signaling processes involved. Importantly, we found that cell survival of MITF- KO cells might be dependent on FAK pathway (*Figure 6g,h*); therefore, this might be the therapeutic vulnerability of MITF-low melanoma cells and can potentially enhance the current treatment options of melanoma.

Upon MITF loss, an EMT-like process has been described to be involved in driving drug resistance in melanoma (*Denecker et al., 2014*; *Caramel et al., 2013*). In addition, the degree of plasticity between EMT and mesenchymal to epithelial transition (MET) has been suggested to lead to high metastatic potential as well as therapeutic resistance (*Stylianou et al., 2019*; *Pastushenko et al., 2018*; *Thompson and Nagaraj, 2018*). Indeed, we observed changes in important EMT markers and regulators such as *ZEB1*, *CDH1* (E-Cadherin), *CDH2* (N-Cadherin), *SNAI2*, and *TGFß1* in the MITF-KO cells (*Figure 4b–e*) as well as in TCGA melanoma samples. Also, the MITF-KO cells showed increased expression of *SOX2*, which is important for neuronal stem cell maintenance and has been suggested to be important for self-renewal of melanoma tumor cells (*Taranova et al., 2006*; *Santini et al., 2014*; *Figure 4b*). Importantly, the effects of MITF on the expression of E-Cadherin, N-Cadherin, and ECM genes (*ITGA2* and *SERPINA3*) is reversible (*Figure 5e–j*). This suggests that MITF enables epithelial to mesenchymal plasticity (EMP) that allows the formation of a hybrid state between EMT and MET to enforce the aggressiveness of melanoma. The binary effects of MITF on the expression of EMT genes may be the molecular mechanism that explains its rheostat activity.

The minimal residual disease (MRD) is a major reason for relapse in cancer. We found that ΔMITF-X6 cells are positively correlated with the gene signature of a population of MRD cells in melanoma tumors as determined by single-cell RNA-seq of human PDX samples and zebrafish melanoma models (*Figure 6—figure supplement 2c–e*; *Figure 2c*; *Rambow et al., 2018*). This gene signature was different from the 'invasive gene signature' that the authors observed in a different set of melanoma cells (*Rambow et al., 2018*). Interestingly, the MRD melanoma cells in zebrafish express little to no MITF protein and have increased expression of ECM genes (*Figure 6—figure supplement 2e*). This suggests that the induced expression of ECM genes and low expression of MITF is one of the markers of MRD in melanoma. Thus, permanently losing MITF reprograms gene expression toward the drug-resistant state, suggesting that MITF-KO cells can be a tool to study drug resistance in melanoma. In the absence of MITF, melanoma cells may become MRD cells by reshaping their ECM, enhancing their attachment to the surface, thus forming quiescent cells which wait for an opportunity to change their phenotype and re-emerge as proliferative melanoma cells. Since melanoma cells can mediate these effects on their own, in the absence of the tumor microenvironment, this suggests that this process is cell-autonomous and under the direction of MITF which instructs the cells to create their own microenvironment.

## Materials and methods

### Cell culture, reagents, and antibodies

SkMel28 (HTB-72) and A375P (CRL-3224) cells were purchased from ATCC and 501Mel melanoma cells were obtained from the lab of Ruth Halaban. The cells were grown in RPMI 1640 medium (#5240025, Gibco) supplemented with 10% FBS (#10270–106, Gibco) at 5% $CO_2$ and 37°C. Cell lines are mycoplasma free and genotypes were confirmed using Short Tandem Repeat (STR) profiling. We made stocks of 5 mM FAK inhibitor (Selleckchem, PF562271) and 5 mM vemurafenib (Selleckchem, S1267) in DMSO and used a dilution of 1 μM final concentration in cell culture media in all drug treatment experiments. The following primary antibodies and their respective dilutions were used in immunofluorescence (IF) and western blot (WB) experiments: MITF (C5) mouse monoclonal (Abcam, #ab12039), 1:2000 (WB), 1:200 (IF); Phospho-Paxillin (Tyr118) rabbit monoclonal (Cell signaling, #2541), 1:000 (WB), 1:100 (IF); Vimentin rabbit monoclonal (Cell signaling, #3932), 1:100 (IF); ERK

(p44/42 MAPK (Erk1/2), CST #9102) 1:1000 (WB); p-ERK (Phospho-p44/42 MAPK (Erk1/2) (Thr202/Tyr204) CST #9101) 1:1000 (WB); E-Cadherin (#610182, BD) 1:5000 (WB), N-Cadherin (#610921, BD) 1:5000 (WB); β-Actin rabbit monoclonal (Cell signaling, #4970), 1:2000 (WB), 1:200 (IF); β-Actin rabbit mouse monoclonal (Millipore, #MAB1501), 1:20000 (WB).

## Generation of MITF-KO cells and validation of mutations using Sanger sequencing and whole genome sequencing

The CRISPR/Cas9 technology was used to generate knock out mutations in the MITF gene in SkMel28 cells. These cells carry the BRAF$^{V600E}$ and p53$^{L145R}$ mutations (*Leroy et al., 2014*). Guide RNAs (gRNAs) were designed targeting exons 2 and 6 of MITF, both of which are common to all isoforms of MITF; exon 2 encodes a conserved domain of unknown function as well as a phosphorylation site, whereas a portion of exon 6 and the entire exon 7 encode the DNA-binding domain of MITF (*Figure 1a*). The gRNAs used were: AGTACCACATACAGCAAGCC (Exon2-gRNA); AGAGTCTGAAGCAAGAGCAC (Exon6-gRNA). The gRNAs were cloned into a gRNA expression vector (Addgene plasmid #43860) using BsmBI restriction digestion. The gRNA vectors were transfected into SkMel28 melanoma cells together with a Cas9 vector (a gift from Keith Joung) using the Fugene HD transfection reagent (#E2312 from Promega) at a 1:2.8 ratio of DNA:Fugene. After transfection, the cells were cultured for 3 days in the presence of 3 µg/ml Blasticidin S (Sigma, stock 2.5 mg/ml) for selection and then serially diluted to generate single cell clones. As a result, we obtained the ΔMITF-X2 cell line from targeting exon 2 of MITF and the ΔMITF-X6 cell line from targeting exon 6. The respective control cell line, termed EV-SKmel28, was generated by transfecting the cells with empty vector Cas9 plasmid.

Genomic DNA was isolated from the MITF knock out cell lines using the following procedures: Cells (~2×105) were trypsinized and spun down and the supernatant was removed. The cell pellet was resuspended in 25 µL of PBS. Then 250 µL Tail buffer (50 mM Tris pH8, 100 mM NaCl, 100 mM EDTA, 1% SDS) containing 2.5 µL of Proteinase K (stock 20 mg/mL) were added to the cell suspension in PBS and incubated at 56°C overnight. Then 50 µL of 5M NaCl were added and mixed on a shaker for 5 min and spun at full speed for 5 min at room temperature. The supernatant was then transferred into a new tube containing 300 µL isopropanol, mixed by inversion and spun in a microfuge for 5 min at full speed. The resulting pellet was washed with 70% ethanol and the pellets airdried at room temperature. Finally, the dried pellets were dissolved in nuclease free water for at least 2 hr at 37°C. The appropriate regions (exons 2 or 6) of MITF were amplified using region-specific primers (MITF-2-Fw: CGTTAGCACAGTGCCTGGTA, MITF-2-Rev: GGGACAAAGGCTGGTAAATG; MITFexon6-fw: GCTTTTGAAAACATGCAAGC, MITFexon6-rev: GGGGATCAATTCTCCCTCTT). The amplified DNA was run on a 1.5% agarose gel, at 70V for 60 min. The bands were cut out of the gel and extracted using Nucleospin Gel and PCR Cleanup Kit (#740609.50 from Macherey Nagel). The purified DNA fragments were cloned into the puc19 plasmid and 10 colonies were picked for each cell line, DNA isolated and sequenced using Sanger sequencing. Whole genome sequencing was performed using total genomic DNA isolated from the EV-SkMel28 as MITF-KO cells using the genomic isolation procedure above. Whole-genome sequencing was performed as described in *Jónsson et al., 2017* using illumina TrueSeq methodology to an average genome wide coverage of 37x. Sequencing results were analyzed using R package CrispRVariant (*Lindsay et al., 2016*) in Bioconductor to quantify mutations introduced in the MITF-KO cell lines.

## Generation of plasmids for stable doxycycline-inducible MITF knockdown and overexpression cell lines

The piggy-bac transposon system was used to generate stable inducible MITF knockdown cell lines. The inducible promoter is a Tetracyclin-On system, which is called reverse tetracyline-transactivator (rtTA). This system allows the regulation of expression by adding tetracycline or doxycycline to the media. We used a piggy-bac transposase vector from Dr. Kazuhiro Murakami (Hokkaido University) (*Magnúsdóttir et al., 2013*). The microRNAs targeting MITF (*Supplementary file 6*) were cloned into the piggy-bac vector downstream of a tetracycline response element (TRE). First, we used BLOCK-iT RNAi designer to design microRNAs targeting MITF (exons 2 and 8 of MITF), including a terminal loop and incomplete sense targeting sequences that are required for the formation of stem loop structures (*Supplementary file 6*). To obtain short double-stranded DNAs with matching BsgI

overhangs, the mature miRNAs were denatured at 95℃, then allowed to cool slowly in a water bath for annealing. Then the piggy-bac vector pPBhCMV1-miR(BsgI)-pA-3 was digested with BsgI (#R05559S, NEB) and the digested vector excised from a DNA agarose gel and the DNA purified. Following this, the annealed primers and purified digested vector were ligated at a 15:1 insert to backbone molar ratio using Instant Sticky-end Ligase Master Mix (M0370S, NEB). A non-targeting control (miR-NTC) was used as a negative control. The ligation products were then transformed to high-competent cells, clones isolated and plasmid DNA sequenced to verify the successful ligation.

For the generation of piggy-bac plasmids containing MITF-M-FLAG-HA and control with only FLAG, we amplified MITF-M cDNA and FLAG sequence from the p3XFLAG-CMV-14 plasmid expressing mouse Mitf-M using the primers listed in *Supplementary file 6* (pB-MITF-M-FLAG-HA), and then introduced it into the piggy-bac vector by restriction digestion with *EcoR* I and *Spe* I.

## Generation of stable doxycycline-inducible MITF knockdown and overexpression cell lines

For generation of stable cells carrying the inducible miR-MITF constructs, 501Mel and SkMel28 cell lines were seeded at 70–80% confluency and then transfected with the following mixture of constructs: py-CAG-pBase, a vector transiently expressing the piggy-bac transposase, MITF targeting plasmids pBhCMV1-miR(MITF-X2)-pA and pPBhCMV1-miR(MITF-X8)-pA encoding miRNA sequences targeting exons 2 and 8 of MITF, and pPB-CAG-rtTA-IRES-Neo, a plasmid which confers neomycin resistance and rtTA. The mixture was in the ratio of 10:5:5:1, respectively. To generate the miR-NTC controls, 501Mel and SkMel28 cells were transfected at a ratio of 10:10:1 with pA-CAG-pBase, pPBhCMV_1-miR(NTC)-pA encoding a non-targeting miRNA and pPB-CAGrtTA-IRES-Neo. For generation of inducible A375P cells carrying the pB-MITF-M-FLAG or a pB-FLAG empty vector, we transfected 70–80% confluent cells with the following plasmids: py-CAG-pBase, pB-MITF-M-FLAG-HA or pB-FLAG-HA, and pPB-CAG-rtTA-IRES-Neo at a 10:10:1 ratio. After 48 hr of transfection, cell lines were subjected to G418 treatment for 2 weeks (0.5 mg/ml, #10131–035, GIBCO) to select for transfected cells.

## RNA isolation, cDNA synthesis, and RT-qPCR

Cells were grown in 6-well culture dishes to 70–80% confluency and RNA was isolated with TRIzol reagent (#15596–026, Ambion), DNase I treated using the RNase free DNase kit (#79254, Qiagen) and re-purified with the RNeasy Mini kit (#74204, Qiagen). The cDNA was generated using High-Capacity cDNA Reverse Transcription Kit (#4368814, Applied Biosystems) using 1 µg of RNA. Primers were designed using NCBI primer blast (*Supplementary file 6*) and qRT-PCR was performed using SensiFAST SYBR Lo-ROX Kit (#BIO-94020, Bioline) on the BIO-RAD CFX38 Real-time PCR machine. The final primer concentration was 0.1 µM and 2 ng of cDNA were used per reaction. Quantitative real-time PCR reactions were performed in triplicates and relative gene expression was calculated using the D-ΔΔCt method (*Livak and Schmittgen, 2001*). The geometric mean of β-actin and human ribosomal protein lateral stalk subunit P0 (RPLP0) was used to normalize gene expression of the target genes. Standard curves were made, and the efficiency calculated using the formula $E = 10 [-1/\text{slope}]$.

## Immunostaining

Cells were seeded on 8-well chamber slides (#354108 from Falcon), grown to 70% confluency and then fixed with 4% paraformaldehyde (PFA) diluted in 1xPBS for 15 min. After washing three times with PBS and blocking with 150 µL blocking buffer (1x PBS + 5% Normal goat serum + 0.3% Triton-X100) for 1 hr at room temperature, cells were stained overnight at 4℃ with the appropriate primary antibodies diluted in antibody staining buffer (1xPBS + 1% BSA + 0,3% Triton-X). The wells were washed three times with PBS and stained for 1 hr at room temperature with the appropriate secondary antibodies, diluted in antibody staining buffer. The wells were washed once with PBS, followed by DAPI staining at a final concentration of 0.5 µg/mL in 1x PBS (1:5000, #D-1306, Life Technologies) and two additional washes with PBS. Subsequently, wells were mounted with Fluoromount-G (Ref 00–495802, ThermoFisher Scientific) and covered with a cover slide. Slides were stored at 4℃ in the dark.

## BrdU assay and FACS analysis

Cells were grown on 6-well plates overnight and treated with a final concentration of 10 mM BrdU for 4 hr. The cells were trypsinized and washed with ice cold PBS and then fixed with 70% ethanol overnight. Next, the cells were centrifuged at 500 g for 10 min and then permeabilized with 2N HCl/Triton X-100 for 30 min followed by neutralization with 0.1 M Na2B4O7.10 H2. Cells were analyzed on a FACS machine (Attune NxT, Thermo fisher scientific) and data were analyzed using FlowJo software.

## IncuCyte live cell imaging

Cells were seeded onto 96-well plates in triplicates supplemented with 200 μL medium with 10% FBS at a density of 2000 cells per well. Images were recorded with the IncuCyte system at 2 hr intervals for a 4-day period. Images were taken with 10x magnification and four images were collected per well. Collected images were then analyzed using the IncuCyte software by measuring cell confluency. Relative confluency was calculated by dividing the confluency at the subsequent hours by the confluency of the initial hour.

## Wound scratch and transwell invasion assay

A total of $2 \times 10^4$ cells were seeded per well of 96-well plate (Nunclon delta surface, Thermo Scientific, #167008) to reach confluent monolayer. Scratches were made with Woundmaker 96 (Essen, Bioscience) and imaging was performed with IncuCyte Live Cell Imaging System (Essen, Bioscience). The recorded images of the scratches were analyzed with IncuCyte software to quantify gap closure. For invasion assay transwell chambers with 8 μm pore size (Thermo Scientific Nunc) were coated with matrigel matrix from Corning (Thermo Scientific). Then cell suspension of $1 \times 10^5$/300 μL in RPMI 1640 supplemented with 0.1% FBS was added to the matrigel coated upper chamber and the medium containing 10% FBS was added to the lower chamber as a chemoattractant. Cell were allowed to invade for 48 hr after which the cells which migrated to the other side of the membrane were fixed with 4% PFA and stained with DAPI. Images were acquired using QImaging (Pecon, software Micro-Manager 1.4.22) with 10x magnification, and the cells were counted using Image J software.

## RNA sequencing and data analysis

We isolated total RNA as described above from EV-SkMel28 and ΔMITF-X6 cell lines and assessed RNA quality using Bioanalyzer. An RNA integrity (RIN) score above eight was used for generating RNA libraries. The mRNA was isolated from total 800 ng RNA using NEBNext Poly(A) mRNA isolation module (E7490, NEB). The RNA was fragmented at 94˚C for 16 min in a thermal cycler. Purified fragmented mRNA was then used to generate cDNA libraries for sequencing using NEBNext Ultra Directional RNA library Kit (E7420S, NEB) following the protocol provided by the manufacturer with these modifications: Adaptors were freshly diluted 10X before use. A total of 15 PCR cycles were used to amplify the library. A total of 8 RNA libraries were prepared with four biological replicates for each cell line including EV-SkMel28 and ΔMITF-X6 cells. Purified RNA sequencing libraries were paired-end sequenced with 30 million reads per sample. Transcript abundance was quantified with Kallisto (*Bray et al., 2016*) and index was built with the GRCh38 reference transcriptome. Differential expression analysis was performed using Sleuth (*Pimentel et al., 2017*) to assess differentially expressed genes between EV-SkMel28 versus ΔMITF-X6. Both likelihood ratio test (LRT) and Wald test were used to model differential expression between ΔMITF-X6 and EV-SkMel28 cells. LRT test is more stringent when estimating differentially expressed genes (DEGs), whereas Wald test gives an estimate for log fold change. Therefore, results from LRT test was intersected with Wald test to get significant DEGs with fold change included. We selected differentially expressed genes with the cut-off of |log2 (foldchange)|≥1 and qval <0.05. Functional enrichment analyses (GO terms and KEGG pathway) were performed using Cluster profiler in the Bioconductor R package using Benjamin-Hochberg test with adjusted p value <0.05 as a cut-off (*Yu et al., 2012*).

Gene set enrichment analysis was performed using GSEA software from the Broad Institute (*Subramanian et al., 2005*). GSEA software was employed with pre-ranked options and gene lists were provided manually to assess enrichment. Differentially expressed genes were ranked combining p-value with log fold change for the input of set enrichment analysis.

## Analysis of human melanoma tumor samples from the Cancer Genome Atlas (TCGA)

The quantified RNA-seq data from 473 melanoma samples were extracted from the Cancer Genome Atlas database using the TCGAbiolinks package in R Bionconductor (*Colaprico et al., 2016*). The lists of MITF$^{low}$ and MITF$^{high}$ samples were generated by sorting the samples based on MITF expression. The 30 tumor samples with the highest MITF expression and 30 tumor samples with the lowest MITF expression were selected for the downstream differential expression analysis built in the TCGAbiolinks package. Principal Component analysis (PCA) plots were generated using normalized count expression of the 200 most significantly differentially expressed genes between MITF$^{low}$ and MITF$^{high}$ samples and EV-SkMel28 and ΔMITF-X6 cells.

## CUT and RUN

To identify direct MITF target genes, we performed anti-MITF Cleavage Under Targets and Release Using Nuclease (CUT and RUN) sequencing in SkMel28 cell lines as described (*Skene and Henikoff, 2017*) with minor modifications. Cells in log-phase culture (approximately 80% confluent) were harvested by cell scraping (Corning), centrifuged at 600 g (Eppendorf centrifuge 5424) and washed twice in calcium-free wash-buffer (20 mM HEPES, pH7.5, 150 mM NaCl, 0.5 mM spermidine and protease inhibitor cocktail, cOmplete Mini, EDTA-free Roche). Pre-activated Concanavalin A-coated magnetic beads (Bangs Laboratories, Inc) were added to cell suspensions (200 K cells) and tubes were incubated at 4°C for 15 min. Antibody buffer (wash-buffer with 2 mM EDTA and 0.03% digitonin) containing anti-MITF (Sigma, HPA003259) or Rabbit IgG (Millipore, 12–370) was added and cells were incubated overnight at 4°C on rotation. The following day, cells were washed in dig-wash buffer (wash buffer containing 0.03% digitonin) and pAG-MNase was added at a concentration of 500 μg/mL. The pAG-MNase enzyme was purified following a previously described protocol (*Meers et al., 2019*). The pAG-MNase reactions were quenched with 2X Stop buffer (340 mM NaCl, 20 mM EDTA, 4 mM EGTA, 0.05% Digitonin, 100 μg/mL RNAse A, 50 μg/mL Glycogen, and 2 pg/mL sonicated yeast spike-in control). Released DNA fragments were Phosphatase K (1 μL/mL, Thermo Fisher Scientific) treated for 1 hr at 50°C and purified by phenol/chloroform-extracted and ethanol-precipitated. CUT-and-RUN experiments were performed in parallel as positive control and fragment sizes analyzed using an 2100 Bioanalyzer (Agilent). All CUT-and-RUN experiments were performed in duplicate.

## Library preparation and data analysis

CUT and RUN libraries were prepared using the KAPA Hyper Prep Kit (Roche). Quality control post-library amplification was conducted using the 2100 Bioanalyzer for fragment analysis. Libraries were pooled to equimolar concentrations and sequenced with paired-end 150 bp reads on an Illumina HiSeq X instrument. Paired-end FastQ files were processed through FastQC (*Andrews, 2010*) for quality control. Reads were trimmed using Trim Galore Version 0.6.3 (Developed by Felix Krueger at the Babraham Institute) and Bowtie2 version 2.1.0 (*Langmead and Salzberg, 2012*) was used to map the reads against the hg19 genome assembly. The mapping parameters were performed as previously described (*Meers et al., 2019*). The accession number for the CUT and RUN sequencing data reported in this paper is GSE153020.

## ChIP-Seq analysis of MITF public dataset

Raw FASTQ files for MITF ChIP-seq were retrieved from GEO archive under the accession numbers GSE50681 and GSE61965 and subsequently mapped to hg19 using bowtie. Peaks were called using MACS, input file was used as control (p value <10e-05) and wig files were generated. Subsequently, wig files were converted to bedgraph using the UCSC tool bigWigToBedGraph with the following command line: wigToBigwig file.wig hg19.chrom.sizes output.bw -clip. The hg19 chromosome size file was downloaded from the UCSC genome browser. We used R package ChIPseeker (*Yu et al., 2015*) for annotation of ChIP-seq peaks to genes, plotting the distribution of peaks around TSS and a fraction of peaks across the genome. For motif analysis, MEMEChIP (*Ma et al., 2014*) was used by extracting DNA sequences corresponding to the peaks that were present in the induced and reduced DEGs of EV-SkMel28 vs. ΔMITF-X6 cells.

## Western blot analysis

A total of 200,000 or 100,000 cells were seeded per well of 12- or 6-well cell culture plates overnight and lysed directly with 1X Laemmli buffer (2% SDS, 5% 2-mercaptoethanol, 10% glycerol, 63 mM Tris-HCl, 0.0025% bromophenol blue, pH 6.8), boiled at 95°C for 10 min and then chilled on ice for 5 min. Lysates were spun down for 1 min at 10,000 rpm, run on 8% SDS-polyacrylamide gels and transferred to 0.2 μm PVDF membranes (#88520 from Thermo Scientific). The membranes were blocked with 5% bovine serum albumin (BSA) in Tris-buffered saline/0.1% Tween 20 (TBS-T) for 1 hr at room temperature, and then incubated overnight (O/N) at 4°C with 5% BSA in TBS-T (20 mM Tris, 150 mM NaCl, 0.05% Tween 20) and the appropriate primary antibodies. Membranes were washed with TBS-T and stained for 1 hr at RT with the appropriate secondary antibodies. The secondary antibodies used were the following: Anti-mouse IgG(H+L) DyLight 800 conjugate (1:15000, #5257) and anti-rabbit IgG(H+L) DyLight 680 conjugate (1:15000, #5366) from Cell Signaling Technologies. The images were captured using Odyssey CLx Imager (LICOR Biosciences).

## Statistical analysis

All statistical tests were performed using GraphPad-Prism, one-way or two-way ANOVA was performed and multiple correction was used as indicated in the figure legends.

## Acknowledgements

This work was supported by grants from the Research Fund of Iceland to ES (184861 and 207067), from the National Institutes of Health to RAC (AR062457), and a grant from the University of Iceland Doctoral Grants Fund to RD. EEP is funded by the MRC HGU Programme (MC_UU_00007/9), European Research Council (ZF-MEL-CHEMBIO-648489), L'Oreal-Melanoma Research Alliance (401181) and Anna-Maria and Stephen Kellen Foundation-Melanoma Research Alliance Team Science Award (#687306). We thank deCODE genetics for their kind assistance with RNA and whole genome sequencing.

## Additional information

### Funding

| Funder | Grant reference number | Author |
|---|---|---|
| Icelandic Centre for Research | 184861 | Eirikur Steingrimsson |
| National Institutes of Health | A2062457 | Robert A Cornell |
| H2020 European Research Council | ZF-MEL-CHEMBIO-648489 | E Elizabeth Patton |
| L'Oreal Melanoma Research Alliance | 401181 | E Elizabeth Patton |
| MRC | MC_UU_00007/9 | E Elizabeth Patton |
| University of Iceland | Doctoral grant fund - HI16120042 | Ramile Dilshat |
| Icelandic Centre for Research | 767-207067 | Eirikur Steingrimsson |
| Anna-Maria and Stephen Kellen Foundation-Melanoma Research Alliance Team Science Award | #687306 | E Elizabeth Patton |

The funders had no role in study design, data collection and interpretation, or the decision to submit the work for publication.

### Author contributions

Ramile Dilshat, Conceptualization, Data curation, Formal analysis, Validation, Investigation, Visualization, Methodology, Writing - original draft, Project administration, Writing - review and editing; Valerie Fock, Conceptualization, Resources, Investigation, Methodology, Writing - review and editing;

Colin Kenny, Data curation, Software, Formal analysis, Methodology, Writing - review and editing; Ilse Gerritsen, Data curation, Software, Formal analysis, Methodology; Romain Maurice Jacques Lasseur, Katrin Möller, Sara Sigurbjörnsdóttir, Methodology; Jana Travnickova, Kritika Kirty, Data curation, Methodology; Ossia M Eichhoff, Philipp Cerny, Data curation, Formal analysis, Methodology, Writing - review and editing; Berglind Ósk Einarsdottir, Phil F Cheng, Data curation; Mitchell Levesque, Resources, Data curation; Robert A Cornell, E Elizabeth Patton, Lionel Larue, Resources, Writing - review and editing; Marie de Tayrac, Erna Magnúsdóttir, Resources, Data curation, Formal analysis, Methodology, Writing - review and editing; Margrét Helga Ögmundsdóttir, Resources, Methodology, Writing - review and editing; Eirikur Steingrimsson, Conceptualization, Resources, Supervision, Investigation, Methodology, Writing - original draft, Project administration, Writing - review and editing

Author ORCIDs
Ramile Dilshat (ID) https://orcid.org/0000-0002-2126-2902
Ossia M Eichhoff (ID) https://orcid.org/0000-0002-3319-1312
Phil F Cheng (ID) http://orcid.org/0000-0003-2940-006X
Robert A Cornell (ID) http://orcid.org/0000-0003-4207-9100
E Elizabeth Patton (ID) http://orcid.org/0000-0002-2570-0834
Erna Magnúsdóttir (ID) https://orcid.org/0000-0002-3369-4390
Eirikur Steingrimsson (ID) https://orcid.org/0000-0001-5826-7486

Decision letter and Author response
Decision letter https://doi.org/10.7554/eLife.63093.sa1
Author response https://doi.org/10.7554/eLife.63093.sa2

## Additional files

### Supplementary files

• Supplementary file 1. List of differentially expressed genes identified in MITF knockdown, knockout, overexpression cell lines vs. respective controls and MITF$^{low}$vs.MITF$^{high}$ melanoma tumors in TCGA.

• Supplementary file 2. MITF CUT-and-RUN targets in SkMel28 cell lines. MITF targets in differentially expressed genes in ΔMITF-X6 vs. SkMel28.

• Supplementary file 3. MITF ChIP-seq targets in the COLO829 cell and bound ECM genes.

• Supplementary file 4. HA-MITF ChIP-seq in the 501Mel cell line and bound ECM genes.

• Supplementary file 5. Differentially expressed extracellular matrix (ECM) genes in MITF knockdown, knockout, overexpression cell lines vs. respective controls and MITF$^{low}$vs. MITF$^{high}$ melanoma tumors in TCGA.

• Supplementary file 6. Primers used in this study.

• Transparent reporting form

### Data availability

MITF CUT&RUN sequencing data have been deposited in GEO under accession codes GSE153020 and the RNA-Seq data discussed in this publication are available under the accession number GSE163646.

The following datasets were generated:

| Author(s) | Year | Dataset title | Dataset URL | Database and Identifier |
|---|---|---|---|---|
| Dilshat R | 2020 | MITF reprograms the extracellular matrix and focal adhesion in melanoma | https://www.ncbi.nlm.nih.gov/geo/query/acc.cgi?acc=GSE163646 | NCBI Gene Expression Omnibus, GSE163646 |
| Kenny C, Cornell RA | 2021 | Cut and Run data of MITF in SKmel28 cells | https://www.ncbi.nlm.nih.gov/geo/query/acc. | NCBI Gene Expression Omnibus, GSE153020 |

cgi?acc=GSE153020

The following previously published datasets were used:

| Author(s) | Year | Dataset title | Dataset URL | Database and Identifier |
|---|---|---|---|---|
| Webster DE, Barajas B, Bussat RT, Yan KJ | 2014 | MITF ChIP-seq in primary melanocyte and melanoma as a function of oncogenic BRAF | https://www.ncbi.nlm.nih.gov/geo/query/acc.cgi?acc=GSE50681 | NCBI Gene Expression Omnibus, GSE50681 |
| Laurette P, Strub T, Koludrovic D, Keime C | 2015 | BRG1 recruitment by transcription factors MITF and SOX10 defines a specific configuration of regulatory elements in the melanocyte lineage (ChIP-seq) | https://www.ncbi.nlm.nih.gov/geo/query/acc.cgi?acc=GSE61965 | NCBI Gene Expression Omnibus, GSE61965 |

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
