## [Decision Letter]

**Acceptance summary:**

In this elegant report, Dilshat and collaborators perform a number of carefully designed experiments in cell lines that show that the transcription factor MITF directly represses the expression of genes associated with the extracellular matrix, focal adhesion and epithelial-mesenchymal transition pathways in human melanoma cells. They then confirm their observations in human tumours from The Cancer Genome Atlas and zebrafish melanomas. These results illustrate the complexity of the MITF regulatory network and are of major importance for understanding resistance of melanoma cells to targeted and immune therapy.

**Decision letter after peer review:**

Thank you for submitting your article "MITF reprograms the extracellular matrix and focal adhesion in melanoma" for consideration by *eLife*. Your article has been reviewed by three peer reviewers, including C Daniela Robles-Espinoza as the Reviewing Editor and Reviewer #1, and the evaluation has been overseen by Richard White as the Senior Editor. The following individuals involved in review of your submission have agreed to reveal their identity: Heinz Arnheiter (Reviewer #2); Stacie Loftus (Reviewer #3).

The reviewers have discussed the reviews with one another and the Reviewing Editor has drafted this decision to help you prepare a revised submission.

In this manuscript, Dilshat and collaborators perform a number of carefully designed experiments in cell lines that show that the transcription factor MITF directly represses the expression of genes associated with the extracellular matrix (ECM), focal adhesion and EMT pathways in human melanoma cells, and confirm their observations in human tumours from TCGA and zebrafish melanomas. The reviewers all agree that the experiments are elegant and the results are sound, and raise only a number of relatively minor issues that would need to be addressed before this manuscript can be published.

Essential revisions:

1) The proliferative/non-invasive and non-proliferative/invasive states of melanoma cells are based on previous gene expression profiling and in vitro experiments. Perhaps due to different experimental conditions, it has also been shown, however, that downregulation of MITF may not lead to large alterations in proliferation and invasion, that nutrient starvation reduces MITF without changing invasive properties, or that normal levels of MITF are compatible with a dual proliferative/invasive phenotype. While it is already difficult to find a common thread to explain these discrepant findings, Dilshat et al. now report that in the absence of MITF or after its downregulation, both proliferation AND invasion are reduced, which is compatible with their finding that a reduction of MITF leads to an increase in focal adhesions. The Discussion would be greatly enhanced if it could be amended to include further insights into why their results are different from those previously published. Can the authors specify if the state they describe represents an additional one to the two mentioned at the beginning of this paragraph? Also, an attempt to reconcile the various findings on MITF high and low states would be welcome in the Discussion section.

2) Regarding the MITF-Χ2 cells, the reviewers have pointed out that additional information regarding the WGS and transcriptomic profile of the CRISPR clones may be necessary to aid the interpretation of experiments involving this clone. One reviewer wrote, "Although the molecular nature of the Χ2 protein would suggest that it is capable of dimerization and DNA binding, Χ2 cells show virtually the same cellular phenotype as MITF-negative X6 cells or occasionally a somewhat milder phenotype than the X6 cells [such as in effects on the expression of certain genes (Figures 3, 4 and 5)]. However, a transcriptomic profile of the Χ2 cells, or a direct analysis of the transcriptional performance of the truncated Χ2 protein, is missing." Can a description of the additional mutations carried by this clone, if any, be included, along with a description of its transcriptional profile?

3) The analysis of MITF expression profiles within the 470 TCGA cutaneous melanoma tumors dataset as described in Figure 3I suggests that analysis was performed on both primary and metastatic tumors. Reviewers suggest that the authors should display or indicate which type of tumors are being analyzed in this analysis. While authors have removed the 130 tumors with "fibroblast marker" expression to assess the MITF high and MITF low tumor profiles, the work would benefit from authors clearly indicating whether the high vs. low MITF profiles are solely reflective of metastatic tumor data, primary tumor data or if this profile holds true regardless of tumor's location.

---

## [Author Response]

Essential revisions:1) The proliferative/non-invasive and non-proliferative/invasive states of melanoma cells are based on previous gene expression profiling and in vitro experiments. Perhaps due to different experimental conditions, it has also been shown, however, that downregulation of MITF may not lead to large alterations in proliferation and invasion, that nutrient starvation reduces MITF without changing invasive properties, or that normal levels of MITF are compatible with a dual proliferative/invasive phenotype. While it is already difficult to find a common thread to explain these discrepant findings, Dilshat et al. now report that in the absence of MITF or after its downregulation, both proliferation AND invasion are reduced, which is compatible with their finding that a reduction of MITF leads to an increase in focal adhesions. The Discussion would be greatly enhanced if it could be amended to include further insights into why their results are different from those previously published. Can the authors specify if the state they describe represents an additional one to the two mentioned at the beginning of this paragraph? Also, an attempt to reconcile the various findings on MITF high and low states would be welcome in the Discussion section.

The rheostat model for the role of MITF in melanoma predicts that high MITF activity promotes proliferation whereas low activity promotes invasion. The reviewers correctly point out that our MITF knockout and knockdown cells do not behave as expected by the model with respect to migratory properties although they do with respect to proliferation. As explained in the Discussion of our manuscript, previous work used siRNA/shRNA to achieve partial and transient depletion of MITF resulting in increased migratory properties of melanoma cells upon MITF reduction (Carreira et al., 2006; Cheli et al., 2012; Javelaud et al., 2011; Bianchi-Smiraglia et al., 2016). Most of the studies used the 501mel and Skmel28 cell lines. Two different pathways involved in migration were shown to be regulated by MITF; *DIAPH1*, a gene implicated in actin polymerization and the guanosine monophosphate reductase (*GMPR*) gene encoding an enzyme involved in regulating intracellular GTP levels (Carreira et al., 2006; Bianchi-Smiraglia et al., 2017). We did not observe changes in *DIAPH1* or *GMPR* expression in our cells or in our analysis of melanoma tumors of TCGA. Importantly, not all studies have resulted in effects on migration, sometimes using the same cells and methods as were used in the previous studies (Falletta et al., 2017 and Vlckova et al., 2018). And interestingly, a recent study showed that reducing MITF alone was not sufficient to mediate invasion; the translation factor eIF2α was needed along with the nutrient sensor ATF4 to mediate invasion of melanoma cells (Falleta et al., 2017). We did not observe changes in eIF2α transcript expression in our cells but did not assess the activity of eIF2α by determining phosphorylation status. Clearly, the experimental system used and the conditions they are performed in play a role in mediating the invasive phenotype.

In our ∆MITF-X6 cell model we have a complete loss of functional MITF. The resulting transcriptional signature was similar to that of the “Rambow neural crest like” cells as determined by single cell RNA-sequencing of melanoma tumors treated with BRAFi (Rambow et al., 2018). These cells are believed to represent minimal residual disease. This gene signature was different from the “invasive gene signature” that the author observed in a different set of melanoma cells (Rambow et al., 2018). Furthermore, comparing our gene signature with that derived from 86 melanoma cells (Hoek et al., 2006), showed that the ∆MITF-X6 cells displayed a more prominent loss of proliferative genes than a gain of invasive genes. Similarly, Rambow et al., 2015, showed that melanoma cell lines (G1, T1, 501Mel, MNT-1, SKMel3) which express high levels of MITF displayed a high proliferation potential but less migratory, invasive and subcutaneous tumour growth capabilities than cell lines with low MITF expression (WM1366, WM793, WM852, LU1205, A375M). The expression of TRIM63 and CAPN3 was over-represented in the cell lines with high MITF expression and knocking down MITF led to a reduction of their expression. Interestingly, knocking down either TRIM63 or CAPN3 enhanced the invasive potential of 501Mel cells. In the ∆MITF-X6 cells we observed reduction only of TRIM63 expression. Thus, the transcriptome of our ∆MITF-X6 cells does not fully recapitulate the gene signature of invasive cells. To emphasize, we did not observe an increased migratory phenotype either using siRNA/miRNA in melanoma cells so this feature is not restricted to our knockout lines.

We could speculate that the migratory phenotype is a transient event and that in order to achieve migration the cells need to be tested at a specific time interval where MITF activity is gradually decreasing. Carmit Levy‘s group has recently shown MITF oscillations upon UV exposure that synchronizes stress response and pigmentation (Malcov-Brog et al., 2018). It is possible that such a mechanism exists, e.g. during melanoblast development where oscillating MITF-expression might ensure that proliferation and migration can both take place. Previous work has suggested oscillations in the dependence of melanoblasts on the receptor tyrosine kinase KIT ( Yoshida et al., 1996; Hou et al., 2000). It is plausible that activation of invasion genes requires discontinuous presence of MITF along with other co-factors at specific time windows. According to the phenotype switching model of melanoma the reversible switch between the proliferative and invasive states is needed for melanoma progression (Arozarena et al., 2019). In our case, the complete loss of MITF might trap the cells in a state where the MITF switch is dead and migration therefore not possible. This would then explain why we do not see the invasive gene signature and no changes in *DIAPH1*, *p27* or *GMPR* expression. However, this would not explain why we do not observe this either using knockdown strategies.

In order to improve the Discussion with respect to this issue we edited the text considerably: The most significant change is the following text:

“Thus, we might speculate that the migratory phenotype is a transient event and that in order to achieve migration the cells need to be tested at a narrow time interval where MITF activity is decreasing. Carmit Levy‘s group has recently shown MITF oscillations upon UV exposure in order to synchronize stress response and pigmentation (Malcov-Brog et al., 2018). [...] Thus, the transcriptome of our ∆MITF-X6 cells does not fully recapitulate the gene signature of invasive cells.”

2) Regarding the MITF-Χ2 cells, the reviewers have pointed out that additional information regarding the WGS and transcriptomic profile of the CRISPR clones may be necessary to aid the interpretation of experiments involving this clone. One reviewer wrote, "Although the molecular nature of the Χ2 protein would suggest that it is capable of dimerization and DNA binding, Χ2 cells show virtually the same cellular phenotype as MITF-negative X6 cells or occasionally a somewhat milder phenotype than the X6 cells [such as in effects on the expression of certain genes (Figures 3, 4 and 5)]. However, a transcriptomic profile of the Χ2 cells, or a direct analysis of the transcriptional performance of the truncated Χ2 protein, is missing." Can a description of the additional mutations carried by this clone, if any, be included, along with a description of its transcriptional profile?

The mutation induced in the ∆MITF-Χ2 line was characterized by both WGS and by PCR-amplification of both genomic DNA and cDNA and subsequent cloning of the resulting fragments into the puc19 vector for Sanger sequencing. The WGS sequencing of the ∆MITF-Χ2 line was included in the original manuscript and we have now added Figure 1—figure supplement 1C showing the Sanger sequencing-results of PCR amplified cDNA fragments. The PCR-seq results are consistent with the WGS results in that we observed an early stop codon at position Y22 (M-MITF) in 5 of 9 clones and a stop codon at position K43 in 4 of 9 clones (Figure 1—figure supplement 1C). We did not observe additional mutations in MITF neither in our WGS sequencing nor in PCR amplified cDNA fragments. The sequence analysis did not reveal any alternative splice products in our mutant clones and searching for them using RT-PCR failed to unravel any such products. Western blotting reveals only the shorter MITF variants in the ∆MITF-Χ2 line that are also observed in wild type but now at a much higher concentration. The size of the smaller MITF variants is estimated at approximately 45 kD (top-most band on Figure 1D) and 40 kD (strongest band on Figure 1D). The difference from the size of the full-length MITF 55 kD protein is 10 and 15 kD, respectively, which corresponds to about 90 and 135 amino acids (excluding possible post-translational modifications that might alter mobility). The fact that these products are detectable using the C5 antibody suggests that they contain the epitope which is located between amino acids 120 and 170 of M-MITF encoded by exons 4 and 5 (Fock et al., 2019). There are several ATGs in the sequence between the initiator MET and the region encoding exon 4, including Met62, Met75, Met77, Met105, Met114, MET116, MET138 and MET144. Some of these have elements of a Kozak-sequence and might represent the alternative products. Mass spec or protein sequencing would be the best way to determine what these products are and at present we do not have the capacity to perform such analysis.

As we were not able to characterize the precise nature of the shorter protein isoforms of MITF that predominate in the Χ2 cells, we hesitated to perform RNA-seq on these cells. However, we performed RT-PCR on RNA isolated from these cells and always observed an intermediate effect on gene expression to that observed in the X6 line, suggesting that this is indeed a hypomorphic mutation.

To provide further detail on the nature of the Χ2 mutation, we have added the following text to the Results section and hope this adds clarity:

“Sanger sequencing of DNA clones containing PCR amplified cDNA fragments from the MITF-gene of ∆MITF-Χ2 cells verified that the ∆MITF-Χ2 cells have the same mutations at similar frequency as were observed in the WGS data (Figure1—figure supplement 1C).” And later in the same paragraph:

“Similarly, neither the WGS nor RT-PCR studies showed evidence for transcripts lacking exon2 from the ∆MITF-Χ2 cDNA indicating that the truncated MITF proteins are most likely products of alternative translation start sites.”

3) The analysis of MITF expression profiles within the 470 TCGA cutaneous melanoma tumors dataset as described in Figure 3I suggests that analysis was performed on both primary and metastatic tumors. Reviewers suggest that the authors should display or indicate which type of tumors are being analyzed in this analysis. While authors have removed the 130 tumors with "fibroblast marker" expression to assess the MITF high and MITF low tumor profiles, the work would benefit from authors clearly indicating whether the high vs. low MITF profiles are solely reflective of metastatic tumor data, primary tumor data or if this profile holds true regardless of tumor's location.

The reviewers are correct that we analysed gene expression in both metastatic and primary tumors before and after withdrawing the fibroblast markers. The melanomas in TCGA are mostly metastatic tumors (371) and primary tumors are much fewer (109). We have now separated the primary tumors from the metastatic ones and then divided them into MITF-high (highest 10%) and low (lowest 10%) tumors. We then performed gene set enrichment analysis and showed that the MITF low tumor samples from both primary and metastatic melanomas showed enrichment for the ECM genes that are affected by MITF loss.

We have edited the paragraph to make these results more clear:

“Next, we analysed whether the collagens that were differentially expressed in the MITF-KD or KO melanoma cell lines were also affected by MITF in melanoma tumors in TCGA. […] The enrichment for ECM genes was observed in both primary and metastatic tumors (Figure 3I).”